# Quasi-hyperbolic momentum and Adam for deep learning

**Jerry Ma**
Facebook AI Research
Menlo Park, CA, USA
maj@fb.com

**Denis Yarats**
Facebook AI Research & New York University
New York, NY, USA
denisy@fb.com

## Abstract

Momentum-based acceleration of stochastic gradient descent (SGD) is widely used in deep learning. We propose the quasi-hyperbolic momentum algorithm (QHM) as an extremely simple alteration of momentum SGD, averaging a plain SGD step with a momentum step. We describe numerous connections to and identities with other algorithms, and we characterize the set of two-state optimization algorithms that QHM can recover. Finally, we propose a QH variant of Adam called QHAdam, and we empirically demonstrate that our algorithms lead to significantly improved training in a variety of settings, including a new state-of-the-art result on WMT16 EN-DE. We hope that these empirical results, combined with the conceptual and practical simplicity of QHM and QHAdam, will spur interest from both practitioners and researchers. Code is immediately available. [1]

## 1 Introduction

Stochastic gradient descent (SGD) serves as the optimizer of choice for many recent advances in deep learning across domains (Krizhevsky et al., 2012; He et al., 2016a; Gehring et al., 2017). SGD for deep learning is typically augmented with either the "heavy ball" momentum technique of Polyak (1964) or the accelerated gradient of Nesterov (1983). In the deterministic setting, these methods provably yield faster convergence in fairly general settings. In the *stochastic* setting, these methods lose many theoretical advantages. However, due to its implicit gradient averaging, momentum can confer the benefit of variance reduction, applying less noisy parameter updates than plain SGD. Recent work has explicitly shown the use of momentum as a variance reducer (Roux et al., 2018).

**Algorithms** Starting with gradient variance reduction as an informal and speculative motivation, we introduce the quasi-hyperbolic momentum (QHM) optimization algorithm in Section 3. Put as simply as possible, QHM's update rule is a weighted average of momentum's and plain SGD's update rule. We later propose a similar variant of Adam (QHAdam) in Section 5.

**Connecting the dots** QHM is simple yet expressive. In Section 4, we connect QHM with plain SGD, momentum, Nesterov's accelerated gradient, PID control algorithms (Recht, 2018; An et al., 2018), synthesized Nesterov variants (Lessard et al., 2016), noise-robust momentum (Cyrus et al., 2018), Triple Momentum (Scoy et al., 2018), and least-squares acceleration of SGD (Kidambi et al., 2018). Such connections yield reciprocal benefits – these algorithms aid in analyzing QHM, and conversely QHM recovers many of these algorithms in a more efficient and conceptually simpler manner. We then characterize the set of optimization algorithms that QHM recovers.

**Practical validation and considerations** In Section 6, we empirically demonstrate that QHM and QHAdam provide superior optimization in a variety of deep learning settings. We provide both comprehensive parameter sweep analyses on smaller models and case studies on large real-world models. We demonstrate improvements on strong (sometimes state-of-the-art) models simply by swapping out the vanilla algorithms with the QH counterpart. Notably, taking the WMT16 EN-DE translation model of Ott et al. (2018), we achieve a 40% improvement in stability, along with a new state-of-the-art result of 29.45 BLEU. We then offer some practical tips for QHM and QHAdam.

---

[1] https://github.com/facebookresearch/qhoptim/

**Miscellany** We provide errata for Kingma & Ba (2015), Recht (2018), and Kidambi et al. (2018). We also offer evidence that momentum often yields negligible improvement over plain SGD.

We emphasize QHM and QHAdam's efficiency and conceptual simplicity. QHM has no extra overhead vs. Nesterov's accelerated gradient, and QHAdam has very little overhead vs. Adam. Also, both algorithms are easily understood as an interpolation between two other well-known algorithms, so they are accessible to practitioners and can be tuned starting with existing practical intuitions. We believe that this contributes strongly to the algorithms' practical promise.

## 2 PRELIMINARIES

We begin with notation and a brief review of stochastic gradient descent (SGD) and momentum.

**Primitives** In this paper, $\theta \in \mathbb{R}^p$ denotes a vector of model parameters. $L(\theta) : \mathbb{R}^p \to \mathbb{R}$ denotes a loss function to be minimized via $\theta$. $\hat{L}(\theta) : \mathbb{R}^p \to \mathbb{R}$ denotes an approximator of the loss function (e.g. over a minibatch). $\nabla L$ denotes the gradient of function $L$. Unless otherwise specified, all vector operations are element-wise. We use $g, a, s, v, w \in \mathbb{R}^p$ as auxiliary buffers, and $g$ is typically the "momentum buffer". $\theta$, $\hat{L}(\cdot)$, and all buffers are subscriptable by $t$, the optimization step.

**Optimization algorithms** We consider optimization algorithms that perform a sequence of steps (indexed by $t$), updating $\theta$ at each step towards minimizing $L(\theta)$. For brevity, we write algorithms as "update rules", which describe the algorithm's behavior during a single step $t$, rather than as full pseudocode. Update rules take this basic form (optionally with one or more auxiliary steps):

$$\theta_{t+1} \leftarrow \theta_t - [\ldots]$$

**Plain SGD** The SGD algorithm, parameterized by learning rate $\alpha \in \mathbb{R}$, uses the update rule:

$$\theta_{t+1} \leftarrow \theta_t - \alpha \cdot \nabla \hat{L}_t(\theta_t)$$

**Momentum** The momentum algorithm, parameterized by $\alpha \in \mathbb{R}$ and $\beta \in \mathbb{R}$, uses the update rule:

$$g_{t+1} \leftarrow \beta \cdot g_t + (1 - \beta) \cdot \nabla \hat{L}_t(\theta_t) \tag{1}$$

$$\theta_{t+1} \leftarrow \theta_t - \alpha \cdot g_{t+1} \tag{2}$$

where $g$ is commonly called the "momentum buffer". Note that $\beta = 0$ recovers plain SGD.

The *exponential discount factor* $\beta$ controls how slowly the momentum buffer is updated. In the stochastic setting, $\beta$ also controls the variance of a normalized momentum buffer. A common rule of thumb for momentum is $\beta = 0.9$ (Ruder, 2016). [2]

In contrast to common formulations of momentum (Polyak, 1964; Sutskever et al., 2013), we normalize, or "dampen", the momentum buffer $g$ by $(1 - \beta)$ in (1). This serves both to remove dependence of the update step magnitude on $\beta$, and to allow the interpretation of $g$ as a weighted average of past gradients (and thus a gradient estimator). Of course, this also shrinks the updates by a factor of $1 - \beta$ vs. common formulations; this is easily reversible with a corresponding increase to $\alpha$.

## 3 ALGORITHM: QUASI-HYPERBOLIC MOMENTUM (QHM)

In this section, we propose and discuss the quasi-hyperbolic momentum (QHM) algorithm.

---
**QHM update rule**

QHM, parameterized by $\alpha \in \mathbb{R}$, $\beta \in \mathbb{R}$, and $\nu \in \mathbb{R}$, uses the update rule:

$$g_{t+1} \leftarrow \beta \cdot g_t + (1 - \beta) \cdot \nabla \hat{L}_t(\theta_t) \tag{3}$$

$$\theta_{t+1} \leftarrow \theta_t - \alpha \left[ (1 - \nu) \cdot \nabla \hat{L}_t(\theta_t) + \nu \cdot g_{t+1} \right] \tag{4}$$

---

Section 7.1 provides a recommended rule of thumb ($\nu = 0.7$ and $\beta = 0.999$).

---
[2]Additionally, Kingma & Ba (2015) recommends $\beta_1 = 0.9$ for Adam, and $\beta_1 = 0.9$ is the default for Adam in both the PyTorch and TensorFlow frameworks (Paszke et al., 2017; Abadi et al., 2015).

**Interpretation**  QHM introduces the *immediate discount factor* $\nu$, encapsulating plain SGD ($\nu = 0$) and momentum ($\nu = 1$). A self-evident interpretation of QHM is as a $\nu$-**weighted average of the momentum update step and the plain SGD update step**.

**QHM vs. momentum**  Comparing (2) and (4), QHM may seem at first glance identical to momentum with discount factor $\nu\beta$. Appendix A.8 analytically demonstrates that this is not the case. We note that the expressive power of QHM intuitively comes from decoupling the momentum buffer's discount factor ($\beta$) from the current gradient's contribution to the update rule ($1 - \nu\beta$). In contrast, momentum tightly couples the discount factor ($\beta$) and the current gradient's contribution ($1 - \beta$).

**Variance reduction**  QHM is originally motivated by an informal and speculative variance reduction analysis; for brevity, we provide the full details in Appendix A. [3] In short, the square bracket term in (4) can be viewed as a gradient estimator (modulo initialization bias). When $\nu = 1$, this is simply the momentum buffer $g_{t+1}$. Increasing $\beta$ decreases the variance of the momentum buffer, but potentially at the cost of making it unusably "stale" (biased). QHM allows for the mitigation of this staleness by upweighting the current, unbiased gradient (i.e. setting $\nu < 1$).

**Efficiency**  QHM, like momentum, requires 1 auxiliary buffer of memory. It also requires 1 in-place scalar-vector multiplication and 3 scaled vector additions per update step.

## 4  CONNECTIONS TO OTHER ALGORITHMS

We now present numerous connections between QHM and other optimization algorithms. The common theme is that QHM recovers almost all of these algorithms, and thus is a highly interpretable and more efficient implementation of these algorithms. The first few subsections present these connections, [4] Table 1 summarizes these connections, and Section 4.5 provides discussion.

### 4.1  NESTEROV'S ACCELERATED GRADIENT

Nesterov (1983)'s accelerated gradient (NAG) can be viewed as a closely related cousin of momentum. In fact, replacing the $g_{t+1}$ term in (2) with $[(1 - \beta) \cdot \nabla\hat{L}_t(\theta_t) + \beta \cdot g_{t+1}]$ yields NAG.

**Connection with QHM**  It follows from (4) that QHM recovers NAG with $\nu = \beta$. This sheds light on the somewhat unintuitive NAG algorithm, providing a natural interpretation of NAG's update rule as a $\beta$-weighted average between momentum and plain SGD.

**Efficiency**  NAG's compute/memory cost is equivalent to that of QHM.

### 4.2  PID CONTROL

Recht (2018) draws a strong connection between gradient-based optimization and PID control. We regurgitate the excellent exposition (with minor modifications) in Appendix B.

**Update rule**  A PID control optimizer, parameterized by $k_P, k_I, k_D \in \mathbb{R}$, uses the update rule:

$$e_t \leftarrow -\nabla\hat{L}_t(\theta_t) \qquad v_t \leftarrow \beta \cdot v_{t-1} + (1 - \beta)(e_t - e_{t-1}) \qquad w_t \leftarrow w_{t-1} + e_t$$
$$\theta_{t+1} \leftarrow \theta_0 + k_P \cdot e_t + k_I \cdot w_t + k_D \cdot v_t$$

**Connection with QHM**  We fully relate QHM and PID in Appendix C.3. To summarize, PID is a superfamily of QHM. Viewing $\beta$ as a constant, QHM imposes a restriction on the ratio between $k_P$ and $k_D$. Viewing $\beta$ as a free variable, however, QHM can recover nearly all PID coefficients.

**Efficiency**  Recht (2018) provides a transformation of variables that reduces the memory cost to 2 auxiliary buffers, and the compute cost to 1 in-place scalar-vector multiplication and 4 scaled vector additions per update step. This is still costlier than QHM.

---

[3]The appendix also sheds some light on the nomenclature, adopted from the hyperbolic discounting work pioneered by Chung & Hernstein (1961), Phelps & Pollak (1968), and Laibson (1997) in consumer choice. We caution that "quasi-hyperbolic" does not directly relate to the geometry of hyperbolas.

[4]Appendix C presents a deeper theoretical treatment of Section 4.2 through Section 4.4, which are largely narrative.

**Alternative PID setting**  In Appendix E, we briefly discuss another PID setting by An et al. (2018) and relate the resulting optimization algorithm to QHM. In short, the setting is degenerate as the P, I, and D terms are linearly dependent. Thus, QHM can recover the resulting PID control optimizer.

### 4.3 Synthesized Nesterov Variants (SNV)

Section 6 of Lessard et al. (2016) describes a "synthesized Nesterov variant" algorithm, which we call "SNV" for convenience. This algorithm is used to analyze and improve optimizer robustness under "relative deterministic noise" (i.e. multiplicative noise of the gradient).

**Update rule**  SNV, parameterized by $\gamma, \beta_1, \beta_2 \in \mathbb{R}$, uses the update rule: [5]

$$\xi_{t+1} \leftarrow \xi_t - \gamma \cdot \nabla \hat{L}_t(\theta_t) + \beta_1(\xi_t - \xi_{t-1})$$
$$\theta_{t+1} \leftarrow \xi_{t+1} + \beta_2(\xi_{t+1} - \xi_t)$$

**Connection with QHM**  We fully relate QHM and SNV in Appendix C.4. To summarize, QHM and SNV recover each other. By extension, QHM recovers the Robust Momentum method, which is a specific parameterization of SNV (Cyrus et al., 2018). Moreover, since Robust Momentum recovers the Triple Momentum of Scoy et al. (2018), QHM also recovers Triple Momentum.

**Efficiency**  SNV is costlier than QHM, requiring 2 auxiliary buffers and 5 scaled vector additions.

### 4.4 AccSGD

Jain et al. (2017) and Kidambi et al. (2018) point out various failures of momentum and NAG in the setting of stochastic least squares optimization. This motivates their proposal of the AccSGD algorithm, which yields faster convergence over momentum and NAG in certain least-squares regression settings. Here, we discuss the formulation of Kidambi et al. (2018).

**Update rule**  AccSGD, parameterized by $\delta > 0$, $\kappa > 1$, $\xi \leq \sqrt{\kappa}$, and $\epsilon < 1$, uses the update rule:

$$\bar{w}_{t+1} \leftarrow \frac{\epsilon^2 \xi}{\kappa} \cdot \bar{w}_t + \left(1 - \frac{\epsilon^2 \xi}{\kappa}\right)\left[w_t - \frac{\kappa \delta}{\epsilon} \cdot \nabla \hat{L}_t(\theta_t)\right]$$

$$\theta_{t+1} = w_{t+1} \leftarrow \frac{\epsilon \xi}{\kappa + \epsilon \xi}\left[w_t - \delta \cdot \nabla \hat{L}_t(\theta_t)\right] + \frac{\kappa}{\kappa + \epsilon \xi} \cdot \bar{w}_{t+1}$$

**Connection with QHM**  We fully relate QHM and AccSGD in Appendix C.5. To summarize, QHM recovers AccSGD. In the reverse direction, AccSGD does not recover QHM; specifically, we disprove the claim in Kidambi et al. (2018) that AccSGD recovers NAG. Since QHM recovers NAG, AccSGD cannot fully recover QHM.

**Efficiency**  AccSGD, like QHM, requires 1 auxiliary buffer. Computationally, AccSGD is costlier, requiring 2 in-place scalar-vector multiplications and 4 scaled vector additions per update step.

### 4.5 Discussion

**Theoretical convergence results**  We note that various convergence results follow simply via these connections. In the deterministic (full-batch) case, since QHM recovers Triple Momentum, QHM also recovers the global linear convergence rate of $1 - 1/\sqrt{\kappa}$ for strongly convex, smooth loss functions. [6]  For first-order methods, this is the fastest known global convergence rate for such functions. In the stochastic (minibatch) case, QHM's recovery of AccSGD gives QHM the same convergence results as in Kidambi et al. (2018)'s least-squares regression setting, of $\mathcal{O}(\sqrt{\kappa} \cdot \log \kappa \cdot \log \frac{1}{\epsilon})$ iterations for $\epsilon$-approximation of the minimal loss.

**Unifying two-state optimization algorithms**  These connections demonstrate that many two-state optimization algorithms are functionally similar or equivalent to each other. However, they are often implemented inefficiently and their parameterizations can be inaccessible to practitioners. QHM yields a highly accessible and efficient version of these algorithms.

---

[5]The learning rate is $\alpha$ in the original paper; we use $\gamma$ to avoid confusion with QHM's $\alpha$.
[6]Here, $\kappa$ is the ratio between the Lipschitz constant of $\nabla L(\cdot)$ and the strong convexity parameter of $L(\cdot)$.

Table 1: Summary of connections between QHM and other optimization algorithms

| ALGORITHM | RELATION [*] | EFF. [†] | BRIEF NOTES |
|---|---|---|---|
| **Plain SGD** | subfamily | better | recovered by QHM with $\nu = 0$ |
| **Momentum** (Polyak, 1964) | subfamily | better | recovered by QHM with $\nu = 1$ |
| **NAG** (Nesterov, 1983) | subfamily | same | recovered by QHM with $\nu = \beta$ |
| **PID** (Recht, 2018) | parent | worse | QHM's $\beta$ restricts PID's $k_P/k_D$ |
| **PID** (An et al., 2018) | bijective | worse | degenerate; either "PI" or "PD" |
| **SNV** (Lessard et al., 2016) | bijective | worse | used in handling multiplicative noise |
| **Robust M.** (Cyrus et al., 2018) | subfamily | worse | SNV w/ convergence guarantees |
| **Triple M.** (Scoy et al., 2018) | subfamily | worse | "fastest" for str. convex, smooth $L(\cdot)$ |
| **AccSGD** (Kidambi et al., 2018) | subfamily | worse | acceleration for least-squares SGD |

[*] **"subfamily"** means that QHM recovers the algorithm but not vice-versa. **"parent"** means that the algorithm recovers QHM but not vice-versa. **"bijective"** means that the algorithms recover each other.
[†] Efficiency (compute and/or memory) vs. QHM.

In Appendix D, we characterize the set of two-state optimization algorithms recoverable by QHM. Our hope here is to provide future work with a routine conversion to QHM so that they may leverage the accessibility and efficiency benefits, as well as the many connections to other algorithms.

**Many-state optimization algorithms** Going beyond a single momentum buffer, it is possible to recover many-state algorithms by linearly combining many momentum buffers (with different discount factors) in the update rule. However, we found in preliminary experiments that using multiple momentum buffers yields negligible value over using a single slow-decaying momentum buffer and setting an appropriate immediate discount – that is, using QHM with high $\beta$ and appropriate $\nu$.

We note that the Aggregated Momentum (AggMo) algorithm (Lucas et al., 2018) precisely performs this linear combination of multiple momentum buffers. While AggMo takes a simple average of the buffers, an extended variant of AggMo allows for other linear combinations. This extended AggMo can be viewed as a many-state generalization of two-state algorithms (including QHM), recovering them when two buffers are used. Appendix H provides a supplemental discussion and empirical comparison of QHM and AggMo, corroborating our preliminary experiments' findings.

## 5 ALGORITHM: QHADAM

The Adam optimizer (Kingma & Ba, 2015) has enabled many compelling results in deep learning (Xu et al., 2015; Vaswani et al., 2017; Yu et al., 2018). We propose to replace both of Adam's moment estimators with quasi-hyperbolic terms, and we name the resulting algorithm QHAdam.

---

**QHAdam update rule**

QHAdam, parameterized by $\alpha, \epsilon \geq 0$, $\beta_1, \beta_2 \in [0, 1)$, and $\nu_1, \nu_2 \in \mathbb{R}$, uses the update rule:

$$g_{t+1} \leftarrow \beta_1 \cdot g_t + (1 - \beta_1) \cdot \nabla \hat{L}_t(\theta_t) \qquad g'_{t+1} \leftarrow \left(1 - \beta_1^{t+1}\right)^{-1} \cdot g_{t+1}$$

$$s_{t+1} \leftarrow \beta_2 \cdot v_t + (1 - \beta_2)(\nabla \hat{L}_t(\theta_t))^2 \qquad s'_{t+1} \leftarrow \left(1 - \beta_2^{t+1}\right)^{-1} \cdot v_{t+1}$$

$$\theta_{t+1} \leftarrow \theta_t - \alpha \left[ \frac{(1 - \nu_1) \cdot \nabla \hat{L}_t(\theta_t) + \nu_1 \cdot g'_{t+1}}{\sqrt{(1 - \nu_2)(\nabla \hat{L}_t(\theta_t))^2 + \nu_2 \cdot s'_{t+1}} + \epsilon} \right]$$

---

Note that only the last expression differs from vanilla Adam. In fact, QHAdam recovers Adam when $\nu_1 = \nu_2 = 1$. Moreover, modulo bias correction, QHAdam recovers RMSProp (Hinton et al., 2012) when $\nu_1 = 0$ and $\nu_2 = 1$, and NAdam (Dozat, 2016) when $\nu_1 = \beta_1$ and $\nu_2 = 1$. We note that Adam has inspired many variants such as AMSGrad (Reddi et al., 2018) and AdamW (Loshchilov & Hutter, 2017), which can be analogously modified.

Table 2: Summary of experimental settings

| SHORT NAME | MODEL | DATASET/TASK | OPTIMIZER |
|---|---|---|---|
| **Logistic-EMNIST-QHM** [PS] | logistic regression | EMNIST digits | QHM |
| **Logistic-EMNIST-QHAdam** [PS] | logistic regression | EMNIST digits | QHAdam |
| **MLP-EMNIST-QHM** [PS] | 3-layer $\tanh$ MLP | EMNIST digits | QHM |
| **MLP-EMNIST-QHAdam** [PS] | 3-layer $\tanh$ MLP | EMNIST digits | QHAdam |
| **RN18-CIFAR10-QHM** [PS] | PreActResNet18 | CIFAR10 | QHM |
| **RN50-ImageNet-QHM** [PS] | ResNet50 | ILSVRC2012 | QHM |
| **RN152-ImageNet-QHM** [CS] | ResNet152 | ILSVRC2012 | QHM |
| **FConvLM-WikiText103-QHM** [CS] | FConvLM | WikiText-103 | QHM |
| **TD3-MuJoCo-QHAdam** [CS] | TD3 | MuJoCo | QHAdam |
| **TF-WMT16ENDE-QHAdam** [CS] | Transformer | WMT16 EN-DE | QHAdam |

[PS] Parameter sweep experiment (Section 6.1).    [CS] Case study (Section 6.2).

**Efficiency**   QHAdam incurs four extra scaled vector additions over Adam.

**Practical notes**   We leave formal convergence analysis to future work. However, a couple of informal points are worth mentioning. Firstly, $\nu_1$ and $\beta_1$ can be reasoned about in a similar manner to QHM's $\nu$ and $\beta$. Secondly, when replacing Adam with QHAdam, setting $\nu_2 = 1$ and $\beta_2$ unchanged is usually reasonable if the original Adam training is stable. Thirdly, when Adam training is *not* stable, it is possible that setting $\nu_2 < 1$ can improve stability by imposing a tighter step size bound – Appendix F elaborates, disproving the step size bound claimed in Kingma & Ba (2015).

## 6   EXPERIMENTS

We perform two categories of experiments: parameter sweeps and case studies. For brevity, all experimental settings are summarized in Table 2 and comprehensively detailed in Appendix I.

### 6.1   PARAMETER SWEEPS

With parameter sweeps, we aim to comprehensively study the various parameterizations of the QH algorithms using relatively small models. We train for 90 epochs with size-64 minibatches. For QHM, we initialize $\alpha = 1$ and decay it 10-fold every 30 epochs. The sweep grid for QHM (encapsulating various parameterizations of plain SGD, momentum, and NAG) is:

$$\nu \in \{0, 0.25, 0.5, 0.6, 0.7, 0.8, 0.9, 0.95, 0.98, 0.99, 0.995, 0.998, 0.999, 0.9995, 1\}$$
$$\beta \in \{0, 0.25, 0.5, 0.6, 0.7, 0.8, 0.9, 0.95, 0.98, 0.99, 0.995, 0.998, 0.999, 0.9995\}$$

For QHAdam, we fix $\alpha = 10^{-3}$, $\epsilon = 10^{-8}$, $\nu_2 = 1$, and $\beta_2 = 0.999$, and sweep over $\nu_1$ and $\beta_1$.

**"Default" $\nu$ and $\beta$ values**   Motivated by the popular momentum/NAG "default" of $\beta = 0.9$, we select a QH "default" of $\nu = 0.7$ and $\beta = 0.999$ based on preliminary experimentation on the MNIST dataset (LeCun, 1998) along with the intuitions from Appendix A. In the following figures, we show these defaults along with the globally optimal parameterizations.

**Results**   Fig. 1 presents selected results of these sweep experiments (full results in Appendix J). Perhaps the most immediate observation is that the QH algorithms improve both training and validation metrics. Even the hardcoded default $\nu = 0.7$ and $\beta = 0.999$ handily outperforms the *optimal* parameterization of NAG or Adam in all settings. In some settings, there remains a large gap between the QH and vanilla algorithms at the end of training. In other settings, the gap shrinks to smaller levels. However, even for these latter settings, the QH algorithm converges much faster, suggesting that a more aggressive learning rate schedule can significantly reduce training time.

**What about plain SGD?**   We note that in most of these experiments, there is little difference between the performance of plain SGD and NAG (particularly when compared to QHM). Although not shown in the figures, there is also little difference between plain SGD and momentum. This indicates that the benefit of momentum and NAG (in the common, unnormalized formulations) comes

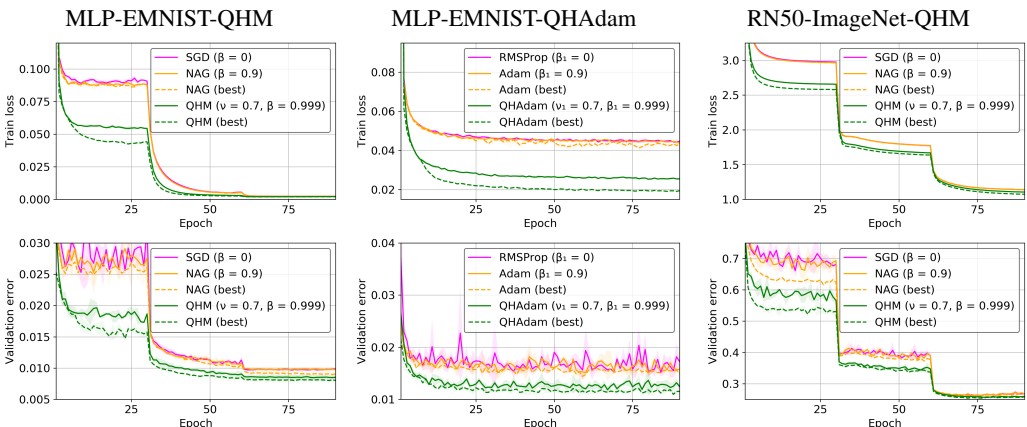

Figure 1: Selected parameter sweep results (full results in Appendix J). Top row shows train loss. Bottom row shows validation error. "Best" refers to the optimal parameterization within the sweep, with respect to the metric. Shaded bands indicate $\pm 1$ standard deviation.

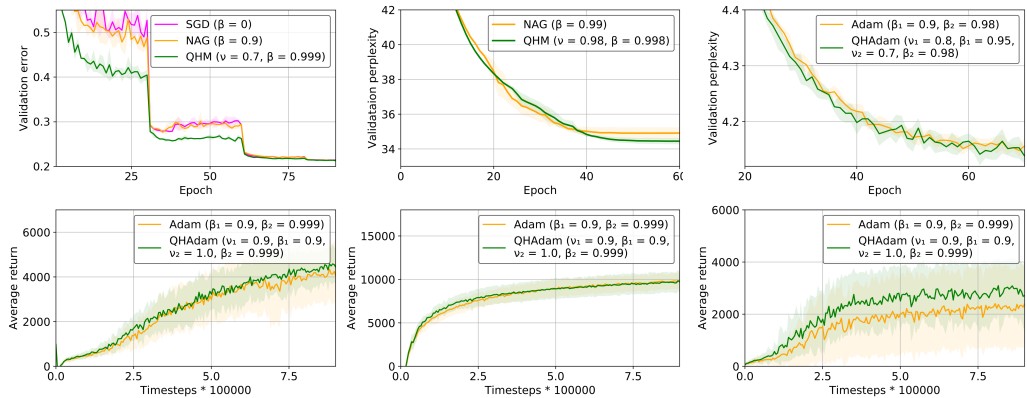

Figure 2: Case study results. Top row: RN152-Imagenet-QHM (left), FConvLM-WikiText103-QHM (center), and TF-WMT16ENDE-QHAdam (right). Bottom row: TD3-MuJoCo-QHAdam Ant (left), HalfCheetah (center), Hopper (right). Shaded bands indicate $\pm 1$ standard deviation.

in large part from the increase in effective step size. We thus suspect that much of the folk wisdom about momentum's benefits for SGD should instead be folk wisdom about using sensible learning rates. In contrast, QHM provides significant benefits *without* changing the effective step size.

## 6.2 CASE STUDIES

With case studies, we apply the QH algorithms to diverse settings, with (currently or recently) state-of-the-art models. Our case studies cover image recognition, language modeling, reinforcement learning, and neural machine translation. Each case study features a baseline setting and a QH setting, which are identical modulo the optimizer used. Results are presented in Fig. 2 and Table 3.

**Image recognition (RN152-ImageNet-QHM)** We train a ResNet152 model (He et al., 2016a) on the ILSVRC2012 dataset (Russakovsky et al., 2015). The baseline setting is nearly identical to the size-256 minibatch baseline in Goyal et al. (2017), using NAG with $\beta = 0.9$ and a decaying learning rate schedule. The QH setting swaps out NAG for QHM, with $\nu = 0.7$ and $\beta = 0.999$. [7]

---

[7]Here, we did not sweep over alternate parameterizations.

Table 3: Case study results

| SHORT NAME | QH | BASELINE |
|---|---|---|
| **RN152-ImageNet-QHM** [ERR] | **0.2128**±0.0005 | 0.2137±0.0011 |
| **FConvLM-WikiText103-QHM** [PPL] | **34.45**±0.17 | 34.92±0.33 |
| **TD3-MuJoCo-QHAdam HalfCheetah** [AR] | **10001.66**±1106.50 | 9829.95±872.29 |
| **TD3-MuJoCo-QHAdam Hopper** [AR] | **2948.14**±1078.70 | 2401.39±1566.00 |
| **TD3-MuJoCo-QHAdam Walker2d** [AR] | 4282.15±707.78 | **4356.95**±860.73 |
| **TD3-MuJoCo-QHAdam Ant** [AR] | **4657.12**±881.03 | 4303.32±1279.86 |
| **TD3-MuJoCo-QHAdam InvPendulum** [AR] | **986.30**±59.70 | 950.27±186.63 |
| **TD3-MuJoCo-QHAdam InvDoublePendulum** [AR] | 8890.59±1941.56 | **9333.49**±20.02 |
| **TD3-MuJoCo-QHAdam Reacher** [AR] | -3.99±0.55 | **-3.98**±0.56 |
| **TF-WMT16ENDE-QHAdam** [BLEU] | **29.45**±0.06 * | 29.17±0.07 |

[ERR] Validation top-1 error rate.    [PPL] Validation perplexity.    [BLEU] Validation BLEU score.
[AR] Average reward.    * State-of-the-art result.

Running 3 seeds, QHM plainly trains much faster than NAG, and QHM converges to a marginally superior validation error as well. [8]

**Language modeling (FConvLM-WikiText103-QHM)** Deep learning for NLP often features "spiky" gradient distributions (e.g. encountering rare words). We train a FConv language model (Dauphin et al., 2016) on the WikiText-103 dataset (Merity et al., 2016). The baseline setting precisely follows the original paper, using NAG with $\beta = 0.99$. The QH setting swaps out NAG for QHM, with $\nu = 0.98$ and $\beta = 0.998$. [7] We suspect that high $\beta$ improves stability in the presense of spiky gradients, and QHM's $\nu$ allows the use of high $\beta$.

Running 10 seeds, QHM outperforms the NAG baseline on validation perplexity by half a point.

**Reinforcement learning (TD3-MuJoCo-QHAdam)** Reinforcement learning presents a challenging task for gradient-based optimization, since the objective $L$ is not stationary. QH algorithms provide a natural way of upweighting the most recent gradient. Here, we apply the TD3 algorithm (Fujimoto et al., 2018) to various MuJoCo environments (Todorov et al., 2012). The baseline precisely follows Fujimoto et al. (2018)'s setup, which uses Adam with $\beta_1 = 0.9$ and $\beta_2 = 0.999$. The QH setting swaps out Adam for QHAdam, with $\nu_1 = 0.9$ and other parameters identical. [9]

Running 10 seeds, QHAdam yields improvements in average reward on four environments out of seven tested, and virtually ties on another.

**Neural machine translation (TF-WMT16ENDE-QHAdam)** Many state-of-the-art neural machine translation (NMT) models are fragile to train. As in language modeling, the gradient distribution is often "spiky"; thus, Adam training often fails to converge due to a very small number of large parameter updates. [10] Here, we empirically demonstrate that QHAdam improves both performance and robustness by using $\nu_2$ to control the maximum per-step update. We train a large transformer model (Vaswani et al., 2017) on the WMT16 English-German dataset. The baseline setting precisely follows the state-of-the-art setup of Ott et al. (2018), using $\beta_1 = 0.9$ and $\beta_2 = 0.98$ for Adam. The QH setting swaps out Adam for QHAdam, with $\nu_1 = 0.8$, $\beta_1 = 0.95$, $\nu_2 = 0.7$, and $\beta_2 = 0.98$. [11]

Running 10 seeds, the Adam baseline explodes on 4 seeds. QHAdam is more robust, converging for all seeds. Ultimately, QHAdam yields a new state-of-the-art-result of **29.45** BLEU. Thus, we improve both the stability and performance of the state-of-the-art with a simple optimizer swap.

---

[8] We also train the model using plain SGD, again finding that plain SGD performs nearly as well as NAG throughout training. Although not shown, plain SGD in fact performs *better* than momentum. The validation loss curves for plain SGD, momentum, and NAG are indistinguishable throughout training, suggesting that momentum/NAG is not needed in Goyal et al. (2017).

[9] Here, we tried higher values of $\beta_1$. Significantly increasing $\beta_1$ was not fruitful for either algorithm.

[10] Refer to Appendix F for a more detailed theoretical treatment.

[11] Here, we tried two other parameterizations (higher $\beta_1$) with marginal success.

## 7 DISCUSSION

### 7.1 PRACTICAL SUGGESTIONS

We offer some practical suggestions for deep learning practitioners, particularly those who default to momentum, NAG, or Adam with $\beta = 0.9$ as a rule of thumb:

- Consider using QHM or QHAdam, instead of momentum, NAG, or Adam.
- While QHM parameters should be tuned when feasible, a decent rule of thumb is to set $\nu = 0.7$ and $\beta = 0.999$. QHAdam parameter selection is somewhat more situational, although as discussed in Section 5, $\nu_2 = 1$ and $\beta_2$ unchanged is usually reasonable when replacing a stable Adam optimizer with QHAdam.
- Be mindful of learning rate differences between (unnormalized) momentum/NAG and QHM. Convert learning rates from the former to the latter via multiplication by $(1 - \beta)^{-1}$. For example, momentum/NAG with $\alpha = 0.1$ and $\beta = 0.9$ should be replaced by QHM with $\alpha = 1$. This conversion is unnecessary for Adam, as it already normalizes all buffers.

### 7.2 FUTURE WORK

This paper has only scratched the surface when it comes to empirical evaluation of QHM and QHAdam. Future work could apply the algorithms to other well-studied tasks and architectures, both to assess the extent of their performance gains in diverse domains, and to further develop insights into hyperparameter choice.

Effective hyperparameter autotuning methods can improve the practicality of any optimization algorithm. Thus, a useful direction for future work is to create an effective $\nu, \beta$ adapter, possibly based on techniques such as YellowFin (Zhang et al., 2017) or via continuous-time optimal control analysis, as in Li et al. (2017). Moreover, learning rate adaptation techniques such as Hypergradient Descent (Baydin et al., 2018) can be applied to both QHM and QHAdam.

Future work could develop convergence results for QHAdam. Convergence results for QHM in a reasonably general stochastic setting would also be appealing, although we are not aware of compelling analogous results for momentum or NAG.

Finally, momentum has been studied in the distributed, asynchronous setting, with some noting that the delays in asynchronous SGD are, in some sense, akin to adding momentum (Mitliagkas et al., 2016). As a result, the optimal momentum constant $\beta$ shrinks as more asynchronous workers are added to optimization. It would be interesting to extend these results to QHM, especially to disentagle the implicit effects of asynchrony on $\nu$ and $\beta$.

### 7.3 CONCLUSION

QHM and QHAdam are computationally cheap, intuitive to interpret, and simple to implement. They can serve as excellent replacements for momentum/NAG and Adam in a variety of settings. In particular, they enable the use of high exponential discount factors (i.e. $\beta$) through the use of immediate discounting (i.e. $\nu$). QHM recovers numerous other algorithms in an efficient and accessible manner. Parameter sweep experiments and case studies demonstrate that the QH algorithms can handily outpace their vanilla counterparts. We hope that practitioners and researchers will find these algorithms both practically useful and interesting as a subject of further study.

### ACKNOWLEDGMENTS

We thank Aaron Defazio, Nicolas Loizou, Yann Olivier, Mark Tygert, and anonymous reviewers and commenters for insightful discussions and valuable suggestions.

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

APPENDICES

**Organization**   This paper's appendices are ordered as follows:

- Appendix A presents a view of momentum and QHM as discounted sums, and provides the original motivation for the development of QHM.

- Appendix B regurgitates Recht (2018)'s excellent exposition of gradient-based optimization as PID control, with minor modifications.

- Appendix C presents analyses of various other algorithms, towards connecting them to QHM.

- Appendix D describes the set of all two-state optimization algorithms recovered by QHM.

- Appendix E briefly discusses a PID control optimization setting by An et al. (2018).

- Appendix F derives a tight upper bound on the updates of Adam and QHAdam (consequently disproving the bound in Kingma & Ba (2015)), then discusses the implications on training stability.

- Appendix G provides miscellaneous derivations that do not cleanly fit in other sections.

- Appendix H provides discussion and an empirical comparison of QHM and AggMo (Lucas et al., 2018).

- Appendix I comprehensively describes the setup of this paper's parameter sweep and case study experiments.

- Appendix J comprehensively presents the results of this paper's parameter sweep experiments.

## A DISCOUNTED SUM ESTIMATORS (ORIGINAL VARIANCE REDUCTION MOTIVATION FOR QHM)

We now provide an interpretation of the momentum buffer as a discounted sum estimator, seeking to motivate the QHM algorithm from a variance reduction perspective.

### A.1 DISCOUNTED SUMS

For a *discount function* $\delta : \mathbb{N}_{\geq 0} \to \mathbb{R}$ and a sequence of vectors $x_{0...t} \in \mathbb{R}^p$, we define a *discounted sum* $\mathrm{DS}_\delta(x_{0...t})$ as:

$$\mathrm{DS}_\delta(x_{0...t}) = \sum_{i=0}^{t} \delta(i) \cdot x_{t-i}$$

When $\sum_{i=0}^{t} \delta(i) = 1$ for all $t \geq 0$, we call this a *discounted sum average*. When $\sum_{i=0}^{\infty} \delta(i) = 1$, we call this a *discounted sum average (modulo initialization bias)*.

### A.2 EXPONENTIAL DISCOUNTING AND EWMA

For $\beta \in (-1, 1)$, we define the *exponential discount function* $\delta_{\mathrm{EXP},\beta}$ as:

$$\delta_{\mathrm{EXP},\beta}(i) = (1 - \beta)\beta^i$$

and the *exponentially weighted moving average* $\mathrm{EWMA}_\beta(x_{0...t})$ as:

$$\mathrm{EWMA}_\beta(x_{0...t}) = \mathrm{DS}_{\delta_{\mathrm{EXP},\beta}}(x_{0...t})$$

$$= (1 - \beta) \cdot \sum_{i=0}^{t} \beta^i \cdot x_{t-i}$$

The EWMA is a discounted sum average (modulo initialization bias), so it can be viewed as an estimator of the expectation of a random variable $x$ if $x_{0...t} \sim x$. Note that the momentum buffer $g_t$ from (1) is precisely an EWMA – specifically, $g_t = \mathrm{EWMA}_\beta(\nabla \hat{L}_{0...t}(\theta_{0...t}))$.

It is well known that the exponential discount function is the only *time-consistent* (commonly "memoryless"), discount function – i.e. for any $i, \tau \geq 0$, the ratio $d(i + \tau)/d(i)$ depends only on $\tau$. This is precisely why the EWMA can be tracked with no auxiliary memory – for example, as in momentum's update rule.

### A.3 EWMA AS VARIANCE REDUCTION

We now provide the following fact about the covariance of the EWMA when $x_{0...t}$ are random variables.

**Fact A.1** (Limit covariance of EWMA). *Assume that $x_{0...t}$ are independent random vectors, each with the covariance matrix $\Sigma$. Then:*

$$\lim_{t \to \infty} \mathrm{Cov}[\mathrm{EWMA}_\beta(x_{0...t})] = \frac{(1 - \beta)^2}{1 - \beta^2} \cdot \Sigma$$

*Proof.* Corollary of Fact A.2. □

This means that arbitrary variance reduction of the EWMA is possible by increasing $\beta$. For example, $\beta = 0.9$ implies that the covariance is reduced to $\frac{1}{19} \cdot \Sigma$, and $\beta = 0.99$ implies that the covariance is reduced to $\frac{1}{199} \cdot \Sigma$.

This provides an intuitive explanation of momentum as a variance reduction technique. Assuming that the momentum buffer is normalized (and thus interpretable as an estimator of the gradient), applying momentum will reduce the variance of the update steps, with higher $\beta$ leading to more variance reduction.

However, the flip side is that higher $\beta$ induces more bias (informally, "staleness") in the momentum buffer with respect to the true gradient, as the momentum buffer becomes extremely slow to update. Thus, the question arises: can we achieve variance reduction while guaranteeing that recent gradients contribute significantly to the update step? For this, we must introduce time-inconsistency.

### A.4 (PURE) HYPERBOLIC DISCOUNTING AND HWMA

Hyperbolic discounting, first proposed by Chung & Hernstein (1961), is the classical time-inconsistent discount function in consumer choice. It is commonly used to model individual behaviors such as impatience. We consider its use in the setting of stochastic optimization (in place of the EWMA buffer of momentum).

For constants $c, k > 0$, we define the *hyperbolic discount function* as: [12]

$$\delta_{\mathrm{H},c,k}(i) = \frac{c}{1 + ki}$$

and the *hyperbolic weighted moving average* $\mathrm{HWMA}_{c,k}(x_{0...t})$ as:

$$\mathrm{HWMA}_{c,k}(x_{0...t}) = \mathrm{DS}_{\delta_{\mathrm{H},c,k}}(x_{0...t})$$

$$= c \cdot \sum_{i=0}^{t} \frac{1}{1 + ki} \cdot x_{t-i}$$

Note that the hyperbolic discount function is time-inconsistent, since:

$$\frac{\delta_{\mathrm{H},c,k}(i + \tau)}{\delta_{\mathrm{H},c,k}(i)} = \frac{1 + ki}{1 + k(i + \tau)}$$

depends on both $i$ and $\tau$.

Unlike the EWMA, the HWMA is not a discounted sum average – in fact, $\sum_{i=0}^{\infty} \delta_{\mathrm{H},c,k}(i) = \infty$ holds regardless of choice of $c$ or $k$. Thus, to use an HWMA of gradients in an optimization algorithm, $c$ (or the learning rate $\alpha$) must be decayed at a logarithmic rate. More concerning, however, is the computational inefficiency of the HWMA; specifically, the sum must be recomputed from scratch at each iteration from all past gradients. This is unacceptable for use in most practical applications.

However, in preliminary stochastic optimization experiments, we did observe a marked benefit of HWMA over EWMA (i.e. momentum), limiting the number of past gradients used for tractability. This indicates that time-inconsistency might be a useful property to have in a stochastic optimizer.

### A.5 QUASI-HYPERBOLIC DISCOUNTING AND QHWMA

Quasi-hyperbolic discounting, proposed by Phelps & Pollak (1968) and popularized in consumer choice by Laibson (1997), seeks to qualitatively approximate the time-inconsistency of hyperbolic discounting by applying a discontinuous "upweighting" of the current step. Its tractability has resulted in much wider adoption in consumer choice vs. pure hyperbolic discounting, and we find that it is also more suited for use in practical optimization.

For constants $\nu \in \mathbb{R}$ and $\beta \in (-1, 1)$, we define the *quasi-hyperbolic discount function* as: [13]

$$\delta_{\mathrm{QH},\nu,\beta}(i) = \begin{cases} 1 - \nu\beta & i = 0 \\ \nu(1 - \beta)\beta^i & i > 0 \end{cases}$$

and the *quasi-hyperbolic weighted moving average* $\mathrm{QHWMA}_{\nu,\beta}(x_{0...t})$ as:

$$\mathrm{QHWMA}_{\nu,\beta}(x_{0...t}) = \mathrm{DS}_{\delta_{\mathrm{QH},\nu,\beta}}(x_{0...t})$$

$$= (1 - \nu\beta) \cdot x_0 + \nu(1 - \beta) \cdot \sum_{i=1}^{t} \beta^i \cdot x_{t-i}$$

$$= (1 - \nu) \cdot x_0 + \nu(1 - \beta) \cdot \sum_{i=0}^{t} \beta^i \cdot x_{t-i} \tag{5}$$

---

[12]Slightly adapted from the original formulation.

[13]Significantly adapted from (but still equivalent to) the original formulation.

The QHWMA, like the EWMA, is a discounted sum average (modulo initialization bias), so it can also be viewed as an estimator under the same assumptions.

When $\nu = 1$, the QHWMA is precisely the EWMA (with identical $\beta$), and the quasi-hyperbolic discount function is precisely the exponential discount function (and thus time-consistent). When $\nu \neq 1$, the quasi-hyperbolic discount function, like the hyperbolic discount function, is time-inconsistent since:

$$\frac{\delta_{\text{QH},\nu,\beta}(i+\tau)}{\delta_{\text{QH},\nu,\beta}(i)} = \begin{cases} \frac{\nu(1-\beta)}{1-\nu\beta}\beta^\tau & i = 0 \\ \beta^\tau & i > 0 \end{cases}$$

depends on both $i$ and $\tau$; specifically, $i = 0$ yields a different ratio than $i > 0$.

Note from (5) that the QHWMA is a $\nu$-weighted average of the EWMA (with identical $\beta$) and $x_0$. This means that the QHWMA can be easily computed online by simply keeping track of the EWMA, thus requiring no additional memory.

### A.6 Variance of QHWMA

We now characterize the variance of a QHWMA using this fact:

**Fact A.2** (Limit covariance of QHWMA). *Assume that $x_{0\ldots t}$ are independent random vectors, each with the covariance matrix $\Sigma$. Then:*

$$\lim_{t\to\infty} \text{Cov}[\text{QHWMA}_{\nu,\beta}(x_{0\ldots t})] = \rho \cdot \Sigma$$

*where $\rho$ is defined as:*

$$\rho = (1-\nu\beta)^2 + \frac{[\nu\beta(1-\beta)]^2}{1-\beta^2}$$

*Proof.* Provided in Appendix G. $\square$

$\rho$ is essentially a scaling factor for the covariance of the QHWMA. It can be verified that $\rho$ decreases (thus inducing variance reduction) with both increasing $\beta$ and increasing $\nu$.

### A.7 Motivating QHM

This leads to our motivation for QHM, which **simply replaces the EWMA momentum buffer with a QHWMA.** Starting with any momentum parameterization ($\nu = 1$ and $\beta \in (0,1)$), $\beta$ can be increased towards variance reduction (i.e. lowering $\rho$). Then, $\nu$ can be decreased to make the QHWMA less biased as a gradient estimator, thus mitigating the aforementioned "staleness" problem. Note, however, that since decreasing $\nu$ will also increase $\rho$, we cannot simply decrease $\nu$ to zero. Specifically, any $\nu < 1$ imposes a tight lower bound of $(1-\nu)^2$ on $\rho$, regardless of choice of $\beta$.

### A.8 Momentum and QHM update rules

For completeness, we explicitly write the update rules for the momentum and QHM algorithms.

**Momentum**   The momentum update rule is:

$$\theta_{t+1} \leftarrow \theta_t - \alpha \left[ (1-\beta) \cdot \sum_{i=0}^{t} \beta^i \cdot \nabla \hat{L}_{t-i}(\theta_{t-i}) \right] \tag{6}$$

which can be efficiently written using an auxiliary buffer $g_t$ as:

$$g_{t+1} \leftarrow \beta \cdot g_t + (1-\beta) \cdot \nabla \hat{L}_t(\theta_t) \qquad \text{(1, revisited from Section 2)}$$
$$\theta_{t+1} \leftarrow \theta_t - \alpha \cdot g_{t+1} \qquad \text{(2, revisited from Section 2)}$$

**QHM**  The QHM update rule is:

$$\theta_{t+1} \leftarrow \theta_t - \alpha \left[ (1 - \nu\beta) \cdot \nabla\hat{L}_t(\theta_t) + \nu(1 - \beta) \cdot \sum_{i=1}^{t} \beta^i \cdot \nabla\hat{L}_{t-i}(\theta_{t-i}) \right] \tag{7}$$

which can be efficiently written using an auxiliary buffer $g_t$ as:

$$g_{t+1} \leftarrow \beta \cdot g_t + (1 - \beta) \cdot \nabla\hat{L}_t(\theta_t) \qquad \text{(3, revisited from Section 3)}$$

$$\theta_{t+1} \leftarrow \theta_t - \alpha \left[ (1 - \nu) \cdot \nabla\hat{L}_t(\theta_t) + \nu \cdot g_{t+1} \right] \qquad \text{(4, revisited from Section 3)}$$

**QHM vs. momentum**  Comparing (2) and (4), QHM may seem at first glance identical to momentum with discount factor $\nu\beta$. However, replacing the $\beta$ in (6) with $\nu\beta$ yields:

$$\theta_{t+1} \leftarrow \theta_t - \alpha \left[ (1 - \nu\beta) \cdot \sum_{i=0}^{t} (\nu\beta)^i \cdot \nabla\hat{L}_{t-i}(\theta_{t-i}) \right]$$

$$\leftarrow \theta_t - \alpha \left[ (1 - \nu\beta) \cdot \nabla\hat{L}_t(\theta_t) + \underbrace{(1 - \nu\beta)}_{\text{differs from (7)}} \cdot \sum_{i=1}^{t} \underbrace{(\nu\beta)^i}_{\text{differs from (7)}} \cdot \nabla\hat{L}_{t-i}(\theta_{t-i}) \right]$$

which plainly differs from (7) – most notably, in the exponential discount factor ($\nu\beta$) for past gradients. Thus, momentum with discount factor $\nu\beta$ does not recover QHM.

## A.9  RELATED WORK IN VARIANCE REDUCTION

Section 4 presents numerous connections to other optimization algorithms that shed light on both deterministic and stochastic convergence properties of QHM. However, we do not formally analyze the convergence properties of QHM from a variance reduction standpoint; this remains future work. Here, we briefly discuss other work in variance reduction.

**Finite sums**  Recently, much effort has been devoted towards reducing the variance of the stochastic gradients used in optimization algorithms. Perhaps the most widely-studied setting is the "finite sum", or offline, stochastic optimization setting. Methods analyzed in the finite-sum setting include SAG (Schmidt et al., 2013), SAGA (Defazio et al., 2014), SVRG (Johnson & Zhang, 2013), FSVRG (Shang et al., 2017), Katyusha (Allen-Zhu, 2017), and others. We do not comment in detail on the finite sum setting due to its limited practical applicability to large-scale deep learning; for a fuller discussion of such methods, see Kidambi et al. (2018).

**Momentum as variance reduction**  Some work in variance reduction has drawn an explicit connection to momentum. For example, Roux et al. (2018) propose a method involving Bayesian updates of gradient estimates, which induces adaptive gradient averaging. The authors note that this method boils down to momentum with an adaptive $\beta$.

## B  PID CONTROLLERS AND OPTIMIZATION

We follow Recht (2018) in describing the connection between PID control and gradient-based optimization.

**Continuous PID**  We slightly adapt the setting from Aaström & Hägglund (1995). $t$ denotes time. There is a setpoint (i.e. target state), $r(t)$, and a process variable (i.e. current state), $y(t)$. The error of the system is defined as $e(t) \overset{\text{def}}{=} r(t) - y(t)$. A "controller" outputs a control signal $u(t)$, usually towards the goal of making the error zero. The controller's choice of $u(t)$ affects $y(t)$ in some unspecified manner.

A PID controller, parameterized by $k_P$, $k_I$, and $k_D$, uses the control function:

$$u(t) = k_P \left[ e(t) \right] + k_I \left[ \int_0^t e(t')dt' \right] + k_D \left[ \frac{de(t)}{dt} \right] \tag{8}$$

Here, the terms in the square brackets are typically referred to as the P, I, and D terms, respectively.

**Discrete approximation**  In discrete time, the setpoint, process variable, and error are trivially discretized as $r_t$, $y_t$, and $e_t \overset{\text{def}}{=} r_t - y_t$, respectively. The I term, which we label $w_t$, is discretized as:

$$w_t \leftarrow w_{t-1} + e_t$$

The D term, which we label $v_t$, could be discretized as $v_t = e_t - e_{t-1}$ (first differences). However, a low-pass filter is often applied to mitigate noise, thus resulting in:

$$v_t \leftarrow \beta v_{t-1} + (1-\beta)(e_t - e_{t-1})$$

We simplify exposition by considering $e_{-1}$, $w_{-1}$, and $v_{-1}$ to be 0.

Finally, the PID control function (8) is trivially discretized as:

$$u_t = k_P \cdot e_t + k_I \cdot w_t + k_D \cdot v_t \tag{9}$$

**Optimization**  Recht (2018) relates optimization to PID control as follows:

$$y_t = \nabla \hat{L}_t(\theta_t) \quad ; \quad r_t = 0 \quad ; \quad e_t \overset{\text{def}}{=} r_t - y_t = -\nabla \hat{L}_t(\theta_t) \quad ; \quad u_t = \theta_{t+1} - \theta_0$$

That is, the process variable is the stochastic gradient, the controller's goal is to make this gradient zero, and the controller achieves this by choosing the next step's model parameters according to the update rule $\theta_{t+1} \leftarrow u_t + \theta_0$. The update rule for a PID control optimizer is thus:

$$e_t \leftarrow -\nabla \hat{L}_t(\theta_t) \qquad v_t \leftarrow \beta v_{t-1} + (1-\beta)(e_t - e_{t-1}) \qquad w_t \leftarrow w_{t-1} + e_t$$
$$\theta_{t+1} \leftarrow \theta_0 + k_P \cdot e_t + k_I \cdot w_t + k_D \cdot v_t$$

Recht demonstrates that PID in this setting encapsulates gradient descent, momentum, and NAG; for example, gradient descent is recovered when $k_P = k_D = 0$ and $k_I = \alpha$.

**Intuition**  Finally, to provide some additional intuition, we can state the following fact about the D term ($v_t$):

**Fact B.1** (D term is gradient and momentum). *$v_t$ can be written as:*

$$v_t = \frac{1 - \beta}{\beta} \left[ -\nabla \hat{L}_t(\theta_t) + (1-\beta) \cdot \sum_{i=0}^{t} \beta^i \cdot \nabla \hat{L}_{t-i}(\theta_{t-i}) \right]$$

*Proof.* Provided in Appendix G. □

Thus, the D term is simply a weighted sum of an EWMA of gradients (i.e. momentum buffer) and the current gradient, and a PID control optimizer's output is simply a weighted sum of the momentum buffer, the current gradient, and the sum of all past gradients.

## C CONNECTIONS TO OTHER ALGORITHMS (IN-DEPTH)

This appendix presents a deeper theoretical treatment of Section 4.2 through Section 4.4, deriving and discussing connections between QHM and various other optimization algorithms.

### C.1 LINEAR OPERATOR ANALYSIS

Along the lines of Lessard et al. (2016), we consider optimizers as linear operators, interrupted by a nonlinear step (the gradient evaluation). In this setting, optimizers have $b$ internal state buffers, which we write as a stacked vector $S_t \in \mathbb{R}^{b \cdot p}$. Optimizers accept the current optimizer state ($S_t$) and gradient ($\nabla \hat{L}_t(\theta_t)$), and they produce the new optimizer state ($S_{t+1}$) and parameters ($\theta_t$) using a square matrix $T \in \mathbb{R}^{(b+2)p \times (b+2)p}$. [14]

**Update rule** For convenience, we impose the restriction that the output $\theta_t$ can only depend on the state $S_t$. Then, for analytical purposes, the optimizer can be written as the following update rule:

$$[S_{t+1} \quad \varnothing \quad \varnothing]^\mathsf{T} \leftarrow T \begin{bmatrix} S_t & \nabla \hat{L}_t(\theta_t) & 0_p \end{bmatrix}^\mathsf{T}$$
$$[\varnothing \quad \varnothing \quad \theta_{t+1}]^\mathsf{T} \leftarrow T \begin{bmatrix} S_{t+1} & 0_p & 0_p \end{bmatrix}^\mathsf{T}$$

where $0_p$ denotes the size-$p$ zero vector, $\varnothing$ denotes throwaway values, and $[v_1 \quad v_2 \quad \ldots]$ denotes vector stacking.

**Coordinate-wise decomposition** Since we only consider optimizers that act coordinate-wise (except for the gradient evaluation), we can write $T$ as the Kronecker product of a coordinate-wise transition matrix $A \in \mathbb{R}^{(b+2) \times (b+2)}$ and the identity matrix $I_p$. That is, $T = A \otimes I_p$.

Then, for $t > 0$, we can write $\theta_t$ in terms of the initial state $S_{0,\{1\ldots b\}}$ and all past gradients, using the last row of various matrix powers of $A$:

$$\theta_t = \sum_{i=1}^{b} \left[ \left( A^{t+1} \right)_{b+2,i} S_{0,i} \right] + \sum_{i=1}^{t} \left[ \left( A^{i+1} \right)_{b+2,b+1} \cdot \nabla \hat{L}_{t-i}(\theta_{t-i}) \right] \tag{10}$$

### C.2 QHM

The internal state of QHM includes two buffers: $g_t$ (momentum buffer) and $\theta_t$ (model parameters).

The transition matrix $T_{\text{QHM}}$, mapping from $\begin{bmatrix} g_t & \theta_t & \nabla \hat{L}_t(\theta_t) & 0_p \end{bmatrix}^\mathsf{T}$ to $\begin{bmatrix} g_{t+1} & \theta_{t+1} & 0_p & \theta_t \end{bmatrix}^\mathsf{T}$, is:

$$A_{\text{QHM}} = \begin{bmatrix} \beta & 0 & 1-\beta & 0 \\ -\alpha\nu\beta & 1 & -\alpha(1-\nu\beta) & 0 \\ 0 & 0 & 0 & 0 \\ 0 & 1 & 0 & 0 \end{bmatrix}$$
$$T_{\text{QHM}} = A_{\text{QHM}} \otimes I_p$$

For $n > 0$, routine computation yields the last row of the $(n+1)$-th matrix power:

$$\left( A_{\text{QHM}}^{n+1} \right)_4 = \begin{bmatrix} -\frac{\alpha\nu\beta(1-\beta^n)}{1-\beta} & 1 & -\alpha(1-\nu\beta^n) & 0 \end{bmatrix}$$

Applying (10), the optimizer state $\theta_t$ can be written as:

$$\theta_t = (A_{\text{QHM}}^{t+1})_{4,1} \cdot g_0 + (A_{\text{QHM}}^{t+1})_{4,2} \cdot \theta_0 + \sum_{i=1}^{t} (A_{\text{QHM}}^{i+1})_{4,3} \cdot \nabla \hat{L}_{t-i}(\theta_{t-i})$$

$$= -\frac{\alpha\nu\beta(1-\beta^t)}{1-\beta} g_0 + \theta_0 - \alpha \cdot \sum_{i=1}^{t} (1-\nu\beta^i) \cdot \nabla \hat{L}_{t-i}(\theta_{t-i})$$

---

[14]Typically, the output would be $\theta_{t+1}$; however, we lag one step so that the parameters can be specified in terms of the current-step state. This is done purely for notational convenience.

In the typical case of $g_0 = 0$, we have:

$$\theta_t = \theta_0 - \alpha \cdot \sum_{i=1}^{t} (1 - \nu\beta^i) \cdot \nabla\hat{L}_{t-i}(\theta_{t-i}) \tag{11}$$

## C.3 PID

Recht (2018) draws a strong connection between gradient-based optimization and PID control. We regurgitate the excellent exposition (with minor modifications) in Appendix B.

**Update rule**   A PID control optimizer, parameterized by $k_P, k_I, k_D \in \mathbb{R}$, uses the update rule:

$$e_t \leftarrow -\nabla\hat{L}_t(\theta_t) \qquad v_t \leftarrow \beta \cdot v_{t-1} + (1-\beta)(e_t - e_{t-1}) \qquad w_t \leftarrow w_{t-1} + e_t$$
$$\theta_{t+1} \leftarrow \theta_0 + k_P \cdot e_t + k_I \cdot w_t + k_D \cdot v_t$$

**Coordinate-wise decomposition**   The internal state of a PID control optimizer includes four buffers: $e_{t-1}$ (P term), $w_{t-1}$ (I term), $v_{t-1}$ (D term), and $\theta_0$ (initial parameters). [15]

The transition matrix $\boldsymbol{T}_{\text{PID}}$, mapping from $\begin{bmatrix} e_{t-1} & w_{t-1} & v_{t-1} & \theta_0 & \nabla\hat{L}_t(\theta_t) & 0_p \end{bmatrix}^{\mathsf{T}}$ to $\begin{bmatrix} e_t & w_t & v_t & \theta_0 & 0_p & \theta_t \end{bmatrix}^{\mathsf{T}}$, is:

$$\boldsymbol{A}_{\text{PID}} = \begin{bmatrix} 0 & 0 & 0 & 0 & -1 & 0 \\ 0 & 1 & 0 & 0 & -1 & 0 \\ -(1-\beta) & 0 & \beta & 0 & -(1-\beta) & 0 \\ 0 & 0 & 0 & 1 & 0 & 0 \\ 0 & 0 & 0 & 0 & 0 & 0 \\ k_P & k_I & k_D & 0 & 0 & 0 \end{bmatrix}$$
$$\boldsymbol{T}_{\text{PID}} = \boldsymbol{A}_{\text{PID}} \otimes \boldsymbol{I}_p$$

For $n > 0$, routine computation yields the last row of the $(n+1)$-th matrix power:

$$\left(\boldsymbol{A}_{\text{PID}}^{n+1}\right)_6 = \begin{bmatrix} -\frac{k_D\beta^{n-1}(1-\beta)}{\beta} & k_I & k_D\beta^n & 1 & \left(\boldsymbol{A}_{\text{PID}}^{n+1}\right)_{6,5} & 0 \end{bmatrix}$$

where:

$$\left(\boldsymbol{A}_{\text{PID}}^{n+1}\right)_{6,5} = \begin{cases} -\left[k_P + k_I + (1-\beta)k_D\right] & n = 1 \\ -\left[k_I - (1-\beta)^2\beta^{n-2}k_D\right] & n > 1 \end{cases}$$

Applying (10), the optimizer state $\theta_t$ can be written as:

$$\theta_t = \left(\boldsymbol{A}_{\text{PID}}^{t+1}\right)_{6,1} \cdot e_{-1} + \left(\boldsymbol{A}_{\text{PID}}^{t+1}\right)_{6,2} \cdot w_{-1} \left(\boldsymbol{A}_{\text{PID}}^{t+1}\right)_{6,3} \cdot v_{-1} + \left(\boldsymbol{A}_{\text{PID}}^{t+1}\right)_{6,4} \cdot \theta_0$$
$$+ \sum_{i=1}^{t} \left(\boldsymbol{A}_{\text{PID}}^{i+1}\right)_{6,5} \cdot \nabla\hat{L}_{t-i}(\theta_{t-i})$$
$$= -\frac{k_D\beta^{t-1}(1-\beta)}{\beta} \cdot e_{-1} + k_I \cdot w_{-1} + k_D\beta^n \cdot v_{-1}$$
$$+ \theta_0 + \sum_{i=1}^{t} \left(\boldsymbol{A}_{\text{PID}}^{i+1}\right)_{6,5} \cdot \nabla\hat{L}_{t-i}(\theta_{t-i})$$

**Relationship with QHM**   In the typical case of $e_{-1} = w_{-1} = v_{-1} = 0_p$, we have:

$$\theta_t = \theta_0 + \sum_{i=1}^{t} \left(\boldsymbol{A}_{\text{PID}}^{i+1}\right)_{6,5} \cdot \nabla\hat{L}_{t-i}(\theta_{t-i})$$

---

[15]The offset of $-1$ in the P, I, and D term subscripts is purely for convenience.

Then, equating with (11), we have that QHM is PID with: [16]

$$k_P = -\frac{\alpha\nu\beta}{1-\beta} \qquad\qquad k_I = \alpha \qquad\qquad k_D = \frac{\alpha\nu\beta^2}{(1-\beta)^2}$$

or that PID is QHM with:

$$\alpha = k_I \qquad\qquad \nu = \frac{k_P^2}{k_D \cdot k_I} \qquad\qquad \beta = \frac{k_D}{k_D - k_P}$$

Viewing $\beta$ as a constant, the following restriction holds on the PID coefficients that QHM can recover:

$$\frac{k_D}{k_P} = -\frac{\beta}{1-\beta}$$

This restriction is looser than those for plain SGD (which has the additional restriction $k_P = k_D = 0$), momentum (which has the additional restriction $k_P/k_I = k_D/k_P$), and NAG (which has the additional restriction $k_P/k_I = \beta k_D/k_P$).

Viewing $\beta$ as a hyperparameter, QHM can recover all PID coefficients except when $k_I = 0$ (i.e. P, D, or PD controller), or $k_P \neq 0 = k_D$ (i.e. PI controller).

To summarize, PID is a superfamily of QHM. Viewing $\beta$ as a constant, QHM imposes a restriction on the ratio between $k_P$ and $k_D$. Viewing $\beta$ as a free variable, however, QHM can recover nearly all PID coefficients.

## C.4 SNV

Section 6 of Lessard et al. (2016) describes a "synthesized Nesterov variant" algorithm, which we call "SNV" for convenience. This algorithm is used to analyze and improve optimizer robustness under "relative deterministic noise" (i.e. multiplicative noise of the gradient). [17]

**Update rule**  SNV, parameterized by $\gamma, \beta_1, \beta_2 \in \mathbb{R}$, uses the update rule: [18]

$$\xi_{t+1} \leftarrow \xi_t - \gamma \cdot \nabla\hat{L}_t(\theta_t) + \beta_1(\xi_t - \xi_{t-1})$$
$$\theta_{t+1} \leftarrow \xi_{t+1} + \beta_2(\xi_{t+1} - \xi_t)$$

**Coordinate-wise decomposition**  The internal state of a SNV optimizer includes two buffers: $\xi_t$ and $\xi_{t-1}$.

The transition matrix $\boldsymbol{T}_{\text{SNV}}$, mapping from $\begin{bmatrix} \xi_t & \xi_{t-1} & \nabla\hat{L}_t(\theta_t) & 0_p \end{bmatrix}^{\mathsf{T}}$ to $\begin{bmatrix} \xi_{t+1} & \xi_t & 0_p & \theta_t \end{bmatrix}^{\mathsf{T}}$, is:

$$\boldsymbol{A}_{\text{SNV}} = \begin{bmatrix} 1+\beta_1 & -\beta_1 & -\gamma & 0 \\ 1 & 0 & 0 & 0 \\ 0 & 0 & 0 & 0 \\ 1+\beta_2 & -\beta_2 & 0 & 0 \end{bmatrix}$$
$$\boldsymbol{T}_{\text{SNV}} = \boldsymbol{A}_{\text{SNV}} \otimes \boldsymbol{I}_p$$

For $n > 0$, routine computation gives us the last row of the $(n+1)$-th matrix power:

$$\left(\boldsymbol{A}_{\text{SNV}}^{n+1}\right)_4 = \begin{bmatrix} \frac{1}{1-\beta_1}(1+\chi_n) & -\frac{1}{1-\beta_1}(\beta_1+\chi_n) & -\frac{\gamma}{\beta_1(1-\beta_1)}(\beta_1+\chi_n) & 0 \end{bmatrix}$$

where:

$$\chi_n = \beta_1^n\left(\beta_2(1-\beta_1) - \beta_1\right)$$

---

[16]This is inconsistent with Recht (2018)'s derivation by a factor of $(1-\beta)$ in $k_P, k_D$ and $(1-\beta)/\beta$ in $k_I$. While $(1-\beta)$ is explainable as the difference between a normalized and unnormalized momentum buffer, we suspect that the extra factor of $\beta$ in $k_I$ is a mistake in the original derivation.

[17]This is similar to the vanishing gradient assumption used in Loizou & Richtárik (2017), which removes the need to consider variance reduction of the gradient.

[18]The learning rate is $\alpha$ in the original paper; we use $\gamma$ to avoid confusion with QHM's $\alpha$.

Applying (10), the optimizer state $\theta_t$ can be written as:

$$
\theta_t = \left( \boldsymbol{T}_{\text{SNV}}^{t+1} \right)_{4,1} \cdot \bar{w}_0 + \left( \boldsymbol{T}_{\text{SNV}}^{t+1} \right)_{4,2} \cdot w_0 + \sum_{i=1}^{t} \left( \boldsymbol{T}_{\text{SNV}}^{i+1} \right)_{4,3} \cdot \nabla \hat{L}_{t-i}(\theta_{t-i})
$$

$$
= \frac{1}{1-\beta_1} (1 + \chi_t) \cdot \xi_0 - \frac{1}{1-\beta_1} (\beta_1 + \chi_t) \cdot \xi_{-1}
$$

$$
- \sum_{i=1}^{t} \frac{\gamma}{\beta_1(1-\beta_1)} (\beta_1 + \chi_i) \cdot \nabla \hat{L}_{t-i}(\theta_{t-i})
$$

**Relationship with QHM**  Initialize $\xi_0 = \xi_{-1} = \theta_0$. The optimizer state $\theta_t$ is:

$$
\theta_t = \theta_0 - \sum_{i=1}^{t} \frac{\gamma}{\beta_1(1-\beta_1)} (\beta_1 + \chi_i) \cdot \nabla \hat{L}_{t-i}(\theta_{t-i})
$$

Then, equating with (11), we have that QHM is SNV with:

$$
\gamma = \alpha(1 - \beta) \qquad\qquad \beta_1 = \beta \qquad\qquad \beta_2 = (1 - \nu)\frac{\beta}{1 - \beta}
$$

or that SNV is QHM with:

$$
\alpha = \frac{\gamma}{1 - \beta_1} \qquad\qquad \nu = 1 - \frac{1 - \beta_1}{\beta_1} \beta_2 \qquad\qquad \beta = \beta_1
$$

To summarize, QHM and SNV recover each other. By extension, QHM recovers the Robust Momentum method, which is a specific parameterization of SNV (Cyrus et al., 2018). Moreover, since Robust Momentum recovers the Triple Momentum of Scoy et al. (2018), QHM also recovers Triple Momentum.

## C.5  ACCSGD

Jain et al. (2017) and Kidambi et al. (2018) point out various failures of momentum and NAG in the setting of stochastic least squares optimization. This motivates their proposal of the AccSGD algorithm, which yields faster convergence over momentum and NAG in certain least-squares regression settings. Here, we discuss the formulation of Kidambi et al. (2018).

**Update rule**  AccSGD, parameterized by $\delta > 0$, $\kappa > 1$, $\xi \leq \sqrt{\kappa}$, and $\epsilon < 1$, uses the update rule:

$$
\bar{w}_{t+1} \leftarrow \frac{\epsilon^2 \xi}{\kappa} \cdot \bar{w}_t + \left( 1 - \frac{\epsilon^2 \xi}{\kappa} \right) \left[ w_t - \frac{\kappa \delta}{\epsilon} \cdot \nabla \hat{L}_t(\theta_t) \right]
$$

$$
\theta_{t+1} = w_{t+1} \leftarrow \frac{\epsilon \xi}{\kappa + \epsilon \xi} \left[ w_t - \delta \cdot \nabla \hat{L}_t(\theta_t) \right] + \frac{\kappa}{\kappa + \epsilon \xi} \cdot \bar{w}_{t+1}
$$

**Coordinate-wise decomposition**  The internal state of an AccSGD optimizer includes two buffers: $\bar{w}_t$ (a buffer) and $w_t$ (the iterate, identical to $\theta_t$).

The transition matrix $\boldsymbol{T}_{\text{AccSGD}}$, mapping from $\begin{bmatrix} \bar{w}_t & w_t & \nabla \hat{L}_t(\theta_t) & 0_p \end{bmatrix}^\mathsf{T}$ to $\begin{bmatrix} \bar{w}_{t+1} & w_{t+1} & 0_p & \theta_t \end{bmatrix}^\mathsf{T}$, is:

$$
\boldsymbol{A}_{\text{AccSGD}} = \begin{bmatrix}
1 - \frac{\epsilon^2 \xi}{\kappa} & \frac{\epsilon^2 \xi}{\kappa} & -\delta \epsilon \xi & 0 \\
\frac{\epsilon \xi}{\kappa + \epsilon \xi} \left( 1 - \frac{\epsilon^2 \xi}{\kappa} \right) & \frac{\kappa + \epsilon^3 \xi^2 / \kappa}{\kappa + \epsilon \xi} & -\delta \frac{\kappa + \epsilon^2 \xi^2}{\kappa + \epsilon \xi} & 0 \\
0 & 0 & 0 & 0 \\
0 & 1 & 0 & 0
\end{bmatrix}
$$

$$
\boldsymbol{T}_{\text{AccSGD}} = \boldsymbol{A}_{\text{AccSGD}} \otimes \boldsymbol{I}_p
$$

For $n > 0$, routine computation gives us the last row of the $(n + 1)$-th matrix power:

$$\left(A_{\text{AccSGD}}^{n+1}\right)_4 = \left[\frac{\kappa - \epsilon^2\xi - (\kappa - \epsilon^2\xi)\chi^n}{\kappa(1+\epsilon)} \quad \frac{\epsilon(\kappa + \epsilon\xi) + (\kappa - \epsilon^2\xi)\chi^n}{\kappa(1+\epsilon)} \quad -\delta\frac{\epsilon(1+\xi) - (\epsilon\xi - 1)\chi^n}{1+\epsilon} \quad 0\right]$$

where:

$$\chi = \frac{\kappa - \epsilon^2\xi}{\kappa + \epsilon\xi}$$

Applying (10), the optimizer state $\theta_t$ can be written as:

$$\theta_t = \left(T_{\text{AccSGD}}^{t+1}\right)_{4,1} \cdot \bar{w}_0 + \left(T_{\text{AccSGD}}^{t+1}\right)_{4,2} \cdot w_0 + \sum_{i=1}^{t}\left(T_{\text{AccSGD}}^{i+1}\right)_{4,3} \cdot \nabla\hat{L}_{t-i}(\theta_{t-i})$$

$$= \frac{\kappa - \epsilon^2\xi - (\kappa - \epsilon^2\xi)\chi^t}{\kappa(1+\epsilon)} \cdot \bar{w}_0 + \frac{\epsilon(\kappa + \epsilon\xi) + (\kappa - \epsilon^2\xi)\chi^t}{\kappa(1+\epsilon)} \cdot w_0$$

$$- \sum_{i=1}^{t}\left(\delta\frac{\epsilon(1+\xi) - (\epsilon\xi - 1)\chi^i}{1+\epsilon}\right) \cdot \nabla\hat{L}_{t-i}(\theta_{t-i})$$

**Relationship with QHM** Fix $\epsilon \in (0, 1)$, and initialize $\bar{w}_0 = w_0 = \theta_0$. The optimizer state $\theta_t$ is:

$$\theta_t = \theta_0 - \sum_{i=1}^{t}\left(\delta\frac{\epsilon(1+\xi) - (\epsilon\xi - 1)\chi^i}{1+\epsilon}\right) \cdot \nabla\hat{L}_{t-i}(\theta_{t-i})$$

Then, equating with (11), we have that QHM is AccSGD with:

$$\delta = \alpha(1 - \nu) \qquad \kappa = \frac{(\beta + \epsilon)(\epsilon\nu + 1)}{(1 - \nu)(1 - \beta)} \qquad \xi = \frac{\epsilon\nu + 1}{\epsilon(1 - \nu)}$$

or that AccSGD is QHM with:

$$\alpha = \frac{\delta\epsilon(1 + \xi)}{1 + \epsilon} \qquad \nu = \frac{\epsilon\xi - 1}{\epsilon(1 + \xi)} \qquad \beta = \frac{\kappa - \epsilon^2\xi}{\kappa + \epsilon\xi}$$

**AccSGD cannot recover NAG** Based on the above analysis, NAG (i.e. $\nu = \beta$) is recovered when $\xi = \frac{1}{2\epsilon}\left(1 - \epsilon + \sqrt{4\kappa + (1 - \epsilon)^2}\right)$.[19] This disproves the claim in Kidambi et al. (2018) that AccSGD recovers NAG when $\xi = \sqrt{\kappa}$. In fact, we demonstrate that AccSGD cannot recover NAG at all. For $\epsilon \in (0, 1)$ and the aforementioned value of $\xi$, we have that $\xi > \sqrt{\kappa}$:

$$\xi = \frac{1}{2\epsilon}\left(1 - \epsilon + \sqrt{4\kappa + (1 - \epsilon)^2}\right) > \frac{1}{2\epsilon}\sqrt{4\kappa + (1 - \epsilon)^2}$$

$$> \frac{2\sqrt{\kappa}}{2\epsilon}$$

$$> \sqrt{\kappa}$$

Since AccSGD requires that $\xi \leq \sqrt{\kappa}$ and that $\epsilon \in (0, 1)$ [20], AccSGD cannot recover NAG.

To summarize, QHM recovers AccSGD. In the reverse direction, AccSGD does not recover QHM; specifically, we disprove the claim in Kidambi et al. (2018) that AccSGD recovers NAG. Since QHM recovers NAG, AccSGD cannot fully recover QHM.

---

[19] Empirical simulations confirm this finding.
[20] The recommended value for $\epsilon$ is 0.7 (Kidambi et al., 2018).

## D  GENERAL TWO-STATE OPTIMIZER

This appendix describes a generic two-state optimizer ("TSO") where one of the states is the iterate ($\theta_t$) and the other is an auxiliary buffer ($a_t$). The optimizer is parameterized by $h, k, l, m, q, z \in \mathbb{R}$, and the update rule is:

$$a_{t+1} \leftarrow h \cdot a_t + k \cdot \theta_t + l \cdot \nabla \hat{L}_t(\theta_t)$$
$$\theta_{t+1} \leftarrow m \cdot a_t + q \cdot \theta_t + z \cdot \nabla \hat{L}_t(\theta_t)$$

We can write this as a transition matrix $\boldsymbol{T}_{\text{TSO}} \in \mathbb{R}^{3 \times 3}$:

$$\boldsymbol{T}_{\text{TSO}} = \begin{bmatrix} h & k & l \\ m & q & z \\ 0 & 0 & 0 \end{bmatrix}$$

To simplify further derivations we diagonalize $\boldsymbol{T}_{\text{TSO}}$ as:

$$\phi = \sqrt{(h-q)^2 + 4km}$$
$$\psi = km - hq$$
$$\boldsymbol{Q}_{\text{TSO}} = \begin{bmatrix} \frac{lq-kz}{\psi} & \frac{h-q-\phi}{2m} & \frac{h-q+\phi}{2m} \\ \frac{hz-lm}{\psi} & 1 & 1 \\ 1 & 0 & 0 \end{bmatrix}$$
$$\boldsymbol{\Lambda}_{\text{TSO}} = \begin{bmatrix} 0 & 0 & 0 \\ 0 & \frac{1}{2}(h+q-\phi) & 0 \\ 0 & 0 & \frac{1}{2}(h+q+\phi) \end{bmatrix}$$
$$\boldsymbol{T}_{\text{TSO}} = \boldsymbol{Q}_{\text{TSO}} \boldsymbol{\Lambda}_{\text{TSO}} \boldsymbol{Q}_{\text{TSO}}^{-1}$$

**Relationship with QHM**  If $\psi \neq 0$, $\phi \neq 0$, $\phi \leq 1$, $\frac{1}{2}(h+q+\phi) = 1$, $h - q + \phi = 0$, and $1 - l = \frac{1}{2}(h+q-\phi)$, then QHM implements the TSO optimizer with:

$$g_0 = -\frac{(2-(h+q-\phi))\psi}{(h+q-\phi)(lq-kz)} \cdot a_0$$
$$\alpha = \frac{1}{2\psi\phi}[(h-q-\phi)(lm-hz) + 2m(lq-kz)]$$
$$\nu = \frac{2m(lq-kz)}{(h-q-\phi)(lm-hz) + 2m(lq-kz)}$$
$$\beta = \frac{1}{2}(h+q-\phi)$$

*Proof.*  We can write down the unrolled TSO update rule for $\theta_t$, as follows:

$$\theta_t \leftarrow (\boldsymbol{T}_{\text{TSO}}^t)_{2,1} \cdot a_0 + (\boldsymbol{T}_{\text{TSO}}^t)_{2,2} \cdot \theta_0 + \sum_{i=0}^{t-1} (\boldsymbol{T}_{\text{TSO}}^{t-i})_{2,3} \cdot \nabla \hat{L}_i(\theta_i)$$

Similarly, for QHM we can define a transition matrix $\boldsymbol{T}_{\text{QHM}} \in \mathbb{R}^{3 \times 3}$ that advances state $\begin{bmatrix} g_t & \theta_t & \nabla \hat{L}_t(\theta_t) \end{bmatrix}$ as:

$$\boldsymbol{T}_{\text{QHM}} = \begin{bmatrix} \beta & 0 & (1-\beta) \\ -\alpha\nu\beta & 1 & -\alpha(1-\nu\beta) \\ 0 & 0 & 0 \end{bmatrix}$$

Thus, the unrolled update rule for QHM takes the following form:

$$\theta_t \leftarrow (\boldsymbol{T}_{\text{QHM}}^t)_{2,1} \cdot g_0 + (\boldsymbol{T}_{\text{QHM}}^t)_{2,2} \cdot \theta_0 + \sum_{i=0}^{t-1} (\boldsymbol{T}_{\text{QHM}}^{t-i})_{2,3} \cdot \nabla \hat{L}_i(\theta_i)$$

Now we match the corresponding coefficients in both of the update rules to establish dependencies:

$$(\boldsymbol{T}_{\text{QHM}}^{t-i})_{2,3} \cdot \nabla \hat{L}_i(\theta_i) = (\boldsymbol{T}_{\text{TSO}}^{t-i})_{2,3} \cdot \nabla \hat{L}_i(\theta_i) \quad \forall i \in [0, t-1]$$

$$(\boldsymbol{T}_{\text{QHM}}^{t})_{2,1} \cdot g_0 + (\boldsymbol{T}_{\text{QHM}}^{t})_{2,2} \cdot \theta_0 = (\boldsymbol{T}_{\text{TSO}}^{t})_{2,1} \cdot a_0 + (\boldsymbol{T}_{\text{TSO}}^{t})_{2,2} \cdot \theta_0$$

By solving the first equation we can establish values for $\alpha$, $\beta$, and $\nu$:

$$(\boldsymbol{T}_{\text{QHM}}^{t-i})_{2,3} = (\boldsymbol{T}_{\text{TSO}}^{t-i})_{2,3} \quad \forall i \in [0, t-1]$$

$$(\boldsymbol{T}_{\text{QHM}}^{t-i})_{2,3} = (\boldsymbol{Q}_{\text{TSO}} \boldsymbol{\Lambda}_{\text{TSO}}^{t-i} \boldsymbol{Q}_{\text{TSO}}^{-1})_{2,3} \quad \forall i \in [0, t-1]$$

$$-\alpha(1 - \nu\beta^{t-i}) = \frac{1}{2\psi\phi}(\boldsymbol{\Lambda}_{\text{TSO}}^{t-i})_{2,2}\left[(h - q + \phi)(lm - hz) + 2m(lq - kz)\right]$$
$$- \frac{1}{2\psi\phi}(\boldsymbol{\Lambda}_{\text{TSO}}^{t-i})_{3,3}\left[(h - q - \phi)(lm - hz) + 2m(lq - kz)\right] \quad \forall i \in [0, t-1]$$

Per our assumption $(\boldsymbol{\Lambda}_{\text{TSO}})_{3,3} = \frac{1}{2}(h + q + \phi) = 1$ and $h - q + \phi = 0$, we can recover the following relationships:

$$\beta = (\boldsymbol{\Lambda}_{\text{TSO}})_{2,2} = \frac{1}{2}(h + q - \phi)$$

$$\alpha = \frac{1}{2\psi\phi}\left[(h - q - \phi)(lm - hz) + 2m(lq - kz)\right]$$

$$\nu = \frac{2m(lq - kz)}{(h - q - \phi)(lm - hz) + 2m(lq - kz)}$$

We can solve the second equation to find $g_0$:

$$(\boldsymbol{T}_{\text{QHM}}^{t})_{2,1} \cdot g_0 + (\boldsymbol{T}_{\text{QHM}}^{t})_{2,2} \cdot \theta_0 = (\boldsymbol{T}_{\text{TSO}}^{t})_{2,1} \cdot a_0 + (\boldsymbol{T}_{\text{TSO}}^{t})_{2,2} \cdot \theta_0$$

$$-\alpha\beta\nu\frac{1 - \beta^t}{1 - \beta} \cdot g_0 + \theta_0 = \frac{m}{\phi}\left[(\boldsymbol{\Lambda}_{\text{TSO}}^{t})_{3,3} - (\boldsymbol{\Lambda}_{\text{TSO}}^{t})_{2,2}\right] \cdot a_0$$
$$+ \frac{1}{2\phi}\left[(\boldsymbol{\Lambda}_{\text{TSO}}^{t})_{2,2}(h - q + \phi) - (\boldsymbol{\Lambda}_{\text{TSO}}^{t})_{3,3}(h - q - \phi)\right] \cdot \theta_0$$

Given that $(\boldsymbol{\Lambda}_{\text{TSO}})_{3,3} = 1$ and $h - q + \phi = 0 \implies h - q - \phi = -2\phi$, we can simplify:

$$-\alpha\beta\nu\frac{1 - \beta^t}{1 - \beta} \cdot g_0 + \theta_0 = \frac{m}{\phi}(1 - \beta^t) \cdot a_0 + \theta_0$$

$$g_0 = -\frac{(1 - \beta)m}{\alpha\beta\nu\phi} \cdot a_0$$

$$g_0 = -\frac{(2 - (h + q - \phi))\psi}{(h + q - \phi)(lq - kz)} \cdot a_0$$

as desired.

$\square$

# E    AN ALTERNATIVE PID SETTING

We very briefly comment on An et al. (2018)'s PID control setting.

**Update rule**   An et al. (2018)'s PID control optimizer, parameterized by $r, k_D, \beta > 0$, uses the following update rule: [21]

$$e_t \leftarrow -\nabla \hat{L}_t(\theta_t) \qquad v_t \leftarrow \beta \cdot v_{t-1} - (1-\beta)(e_t - e_{t-1}) \qquad w_t \leftarrow \beta \cdot w_{t-1} + r \cdot e_t$$

$$u_t \leftarrow w_t + k_D \cdot v_t$$

$$\theta_{t+1} \leftarrow \theta_t + u_t$$

**Discussion**   This setting departs somewhat from typical PID control, in that the signal $u_t$ controls the derivative of the controller's output (i.e. $\theta_{t+1} - \theta_t$) rather than the output itself (i.e. $\theta_{t+1} - \theta_0$). To avoid parameter blowup, this formulation necessitates the addition of exponential decay to the I term, with discount factor $\beta$. [22]

The I term thus becomes the momentum buffer. However, recall from Fact B.1 that the D term is a weighted sum of the momentum buffer and the P term. It follows that the D term is a weighted sum of the P and I terms, and that this setting is degenerate (either "PI" or "PD").

As a consequence, the proposed PID algorithm of An et al. (2018) is less expressive than that of Recht (2018). Specifically, applying Fact B.1 demonstrates a mapping into QHM:

$$\alpha = \frac{r}{1-\beta} \qquad\qquad \nu = 1 - \frac{k_D(1-\beta)^2}{r\beta} \qquad\qquad \beta = \text{unchanged}$$

**Efficiency**   This PID control optimizer is costlier than QHM. It requires 2 auxiliary buffers of memory. Computationally, it requires 2 in-place scalar-vector multiplications and 5 scaled vector additions per update step.

---

[21]The exponential discount is $\alpha$ in the original paper; we use $\beta$ to avoid confusion.
[22]One final oddity is that the D term ($v_t$) calculates the negation of the derivative.

## F  (QH)ADAM'S UPDATE BOUND

This appendix elaborates on Adam and QHAdam's stability properties through the lens of a step size upper bound.

It is well known that the training process for deep learning models can often "explode" due to a very small number of large parameter updates. With Adam, these large updates can occur if there exist parameters whose stochastic gradients are almost always near zero but incur rare "spikes". [23]. This is because the square root of the second moment estimate, used in normalizing the gradient for the update step, will be far below the magnitude of these spikes.

There are three main ways to address this instability:

- Firstly, one can simply decrease the learning rate $\alpha$. However, this may be undesirable due to slower training.

- Secondly, one can increase the $\epsilon$ hyperparameter. However, the appropriate setting of $\epsilon$ depends on the exact magnitudes of these gradient spikes, which is often unknown. Setting $\epsilon$ too high effectively turns Adam into SGD. Thus, setting $\epsilon$ often reduces to guesswork.

- Thirdly, one can clip gradients. However, the appropriate magnitude of the gradient clipping also depends on the stochastic gradient distribution. Thus, this solution also involves a fair amount of guesswork.

However, Adam does provide a useful guarantee – unlike SGD, Adam has an upper bound on the per-step update (Kingma & Ba, 2015). This upper bound is independent of the gradient distribution (or even temporal correlation), depending only on the hyperparameters $\alpha$, $\beta_1$, and $\beta_2$. Thus, no matter the gradient distribution, Adam will restrict the magnitude of the per-step updates to some known constant. Kingma & Ba (2015) intuitively describe this bound as "establishing a trust region around the current parameter value".

We show that the step size upper bound claimed in Section 2.1 of Kingma & Ba (2015) is incorrect, by providing the correct tight bound for both Adam and QHAdam. We then demonstrate that with QHAdam, one can lower the maximum per-step update (and thus improve stability) simply by lowering $\nu_2$ to be below 1.

### F.1  SETTING

We make two simplifications. Firstly, we fix $\epsilon = 0$. [24] Secondly, we remove the bias correction of the moment estimators (i.e. we use $g'_{t+1} \leftarrow g_{t+1}$ and $s'_{t+1} \leftarrow s_{t+1}$).

In this setting, QHAdam applies the following update rule:

$$\theta_{t+1} \to \theta_t - \alpha \cdot \frac{\tilde{g}_{t+1}}{\sqrt{\tilde{s}_{t+1}}}$$

where:

$$\tilde{g}_{t+1} = (1 - \nu_1) \cdot \nabla \hat{L}_t(\theta_t) + \nu_1(1 - \beta_1) \cdot \sum_{i=0}^{t} \beta_1^i \cdot \nabla \hat{L}_{t-i}(\theta_{t-i}) \tag{12}$$

$$\tilde{s}_{t+1} = (1 - \nu_2)(\nabla \hat{L}_t(\theta_t))^2 + \nu_2(1 - \beta_2) \cdot \sum_{i=0}^{t} \beta_2^i (\nabla \hat{L}_{t-i}(\theta_{t-i}))^2 \tag{13}$$

### F.2  IMPLICIT UPDATE BOUND

We now bound QHAdam's update (before scaling by $\alpha$) by a constant dependent only on $\beta_1$, $\beta_2$, $\nu_1$, and $\nu_2$:

---

[23]Somewhat more concretely, a stochastic gradient is "spiky" if its distribution has extremely large higher-order cumulants relative to its typical magnitudes.

[24]The analysis with $\epsilon$ free follows the same approach but is somewhat messier.

**Fact F.1** (QHAdam tight upper bound on update). *Assume that $\tilde{s}_{t+1}$ is nonzero at each coordinate and that $0 < \beta_1 < \sqrt{\beta_2} < 1$. Then, the following per-coordinate tight upper bound holds:*

$$\left| \frac{\tilde{g}_{t+1}}{\sqrt{\tilde{s}_{t+1}}} \right|_\infty \leq \sqrt{\frac{(1 - \nu_1 \beta_1)^2}{1 - \nu_2 \beta_2} + \frac{[\nu_1 \beta_1 (1 - \beta_1)]^2 \left[ 1 - \left( \frac{\beta_1^2}{\beta_2} \right)^t \right]}{\nu_2 (1 - \beta_2)(\beta_2 - \beta_1^2)}}$$

*Proof.* Firstly and without loss of generality, we can treat the gradients as single coordinates $x_i \in \mathbb{R}$. That is, $x_i = \nabla \hat{L}_i(\theta_i) \in \mathbb{R}$ for $i \in [0, t]$.

We perform the following simplification of (12) and (13):

$$\tilde{g}_{t+1} = (1 - \nu_1 \beta_1) \cdot x_t + \nu_1 (1 - \beta_1) \cdot \sum_{i=1}^{t} \beta_1^i \cdot x_{t-i} \tag{14}$$

$$\tilde{s}_{t+1} = (1 - \nu_2 \beta_2) \cdot x_t^2 + \nu_2 (1 - \beta_2) \cdot \sum_{i=1}^{t} \beta_2^i \cdot x_{t-i}^2 \tag{15}$$

We now wish to find the values of $x_i$ that maximize $\frac{\tilde{g}_{t+1}^2}{\tilde{s}_{t+1}}$. Applying (14) and (15), these values are characterized by the following first-order conditions:

$$x_i = \begin{cases} \frac{\tilde{s}_{t+1}}{\tilde{g}_{t+1}} \left[ \frac{\nu_1 (1 - \beta_1) \beta_1^i}{\nu_2 (1 - \beta_2) \beta_2^i} \right] & i < t \\ \frac{\tilde{s}_{t+1}}{\tilde{g}_{t+1}} \left[ \frac{1 - \nu_1 \beta_1}{1 - \nu_2 \beta_2} \right] & i = t \end{cases} \tag{16}$$

Since the quantity $\frac{\tilde{g}_{t+1}^2}{\tilde{s}_{t+1}}$ is invariant to scalar multiplication of all $x_i$, we can simplify (16) to:

$$x_i = \begin{cases} \frac{\nu_1 (1 - \beta_1) \beta_1^i}{\nu_2 (1 - \beta_2) \beta_2^i} & i < t \\ \frac{1 - \nu_1 \beta_1}{1 - \nu_2 \beta_2} & i = t \end{cases} \tag{17}$$

Plugging the values of $x_i$ from (17) into (14) and (15) yields:

$$\max_{x_{0 \ldots t}} \frac{\tilde{g}_{t+1}^2}{\tilde{s}_{t+1}} = \frac{(1 - \nu_1 \beta_1)^2}{1 - \nu_2 \beta_2} + \frac{[\nu_1 \beta_1 (1 - \beta_1)]^2 \left[ 1 - \left( \frac{\beta_1^2}{\beta_2} \right)^t \right]}{\nu_2 (1 - \beta_2)(\beta_2 - \beta_1^2)}$$

The desired result follows immediately. $\qquad \square$

**Limit case** Consider the limit case of $t \to \infty$. Then, the bound in Fact F.1 simplifies to:

$$\left| \frac{\tilde{g}_{t+1}}{\sqrt{\tilde{s}_{t+1}}} \right|_\infty \leq \sqrt{\frac{(1 - \nu_1 \beta_1)^2}{1 - \nu_2 \beta_2} + \frac{[\nu_1 \beta_1 (1 - \beta_1)]^2}{\nu_2 (1 - \beta_2)(\beta_2 - \beta_1^2)}} \tag{18}$$

For vanilla Adam (i.e. $\nu_1 = \nu_2 = 1$), (18) simplifies further to:

$$\left| \frac{\tilde{g}_{t+1}}{\sqrt{\tilde{s}_{t+1}}} \right|_\infty \leq (1 - \beta_1) \sqrt{\frac{\beta_2}{(1 - \beta_2)(\beta_2 - \beta_1^2)}} \tag{19}$$

Note that since the bound in (19) is tight, this result contradicts the claim in Section 2.1 of Kingma & Ba (2015) that Adam's per-coordinate step size is bounded above by $\alpha \cdot \max\{1, (1 - \beta_1)/\sqrt{1 - \beta_2}\}$.[25] In the following discussion, we use the correct bounds from (18) and (19).

---

[25]The difference can be rather large – for example, for the recommended Adam parameters of $\beta_1 = 0.9$ and $\beta_2 = 0.999$, Kingma & Ba (2015) claim an upper bound of $\lesssim 3.16 \cdot \alpha$, while (19) implies a tight bound of $\lesssim 7.27 \cdot \alpha$.

### F.3 DISCUSSION

The recommended vanilla Adam setting of $\beta_2 = 0.999$ in Kingma & Ba (2015) makes the right-hand side of (19) to be large, and various work has employed Adam with a significantly lower $\beta_2$; e.g. 0.98 (Vaswani et al., 2017; Ott et al., 2018). [26] Decreasing $\beta_2$ is undesirable, often slowing down training. [27] Moving from Adam to QHAdam, an alternative solution is to decrease $\nu_2$ to be below 1. This decreases the right-hand side of (18), up to a point, and thus imposes a tighter constraint on the magnitudes of updates than the vanilla Adam setting of $\nu_2 = 1$. Fig. 3 shows an example of this phenomenon using a fixed $\nu_1$, $\beta_1$, and $\beta_2$.

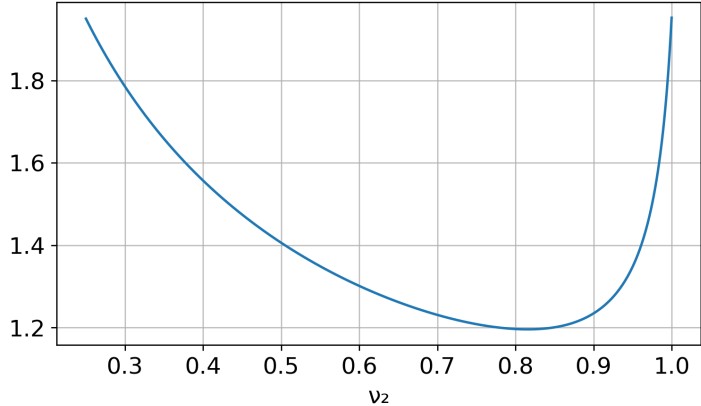

Figure 3: Bound from (18), fixing $\nu_1 = 0.8$, $\beta_1 = 0.95$, and $\beta_2 = 0.98$, and varying $\nu_2$.

---

[26]We performed experiments on these models indicating that increasing $\beta_2$ far beyond 0.98 led to training explosion. We suspect that these instability issues are especially prevalent in settings with rare inputs or labels, such as machine translation.

[27]In proposing the AdamNC algorithm, Reddi et al. (2018) suggests that $\beta_2$ should be high to capture a sufficiently long history of past gradients.

## G  MISCELLANEOUS DERIVATIONS

This appendix provides miscellaneous derivations that do not cleanly fit elsewhere.

**Fact A.2** (Limit covariance of QHWMA). *Assume that $x_{0...t}$ are independent random vectors, each with the covariance matrix $\Sigma$. Then:*

$$\lim_{t \to \infty} \text{Cov}[\text{QHWMA}_{\nu,\beta}(x_{0...t})] = \rho \cdot \Sigma$$

*where $\rho$ is defined as:*

$$\rho = (1 - \nu\beta)^2 + \frac{[\nu\beta(1 - \beta)]^2}{1 - \beta^2}$$

*Proof.* Due to the independence assumption, the covariance matrix of the QHWMA for $t > 0$ is simply:

$$\text{Cov}[\text{QHWMA}_{\nu,\beta}(x_{0...t})] = \left[ \sum_{i=0}^{t} \delta^2_{\text{QH},\nu,\beta}(i) \right] \Sigma$$

$$= \left[ (1 - \nu\beta)^2 + \sum_{i=1}^{t} \left( \nu(1 - \beta)\beta^i \right)^2 \right] \Sigma$$

$$= \left[ (1 - \nu\beta)^2 + (\nu(1 - \beta))^2 \sum_{i=1}^{t} \beta^{2i} \right] \Sigma$$

$$= \left[ (1 - \nu\beta)^2 + \frac{\nu^2(1 - \beta)^2 \beta^2 (1 - \beta^{2t})}{1 - \beta^2} \right] \Sigma$$

The desired result follows immediately. $\qquad\square$

**Fact B.1** (D term is gradient and momentum). *$v_t$ can be written as:*

$$v_t = \frac{1 - \beta}{\beta} \left[ -\nabla \hat{L}_t(\theta_t) + (1 - \beta) \cdot \sum_{i=0}^{t} \beta^i \cdot \nabla \hat{L}_{t-i}(\theta_{t-i}) \right]$$

*Proof.* We expand $v_t$ as follows, recalling that $v_{-1} = 0$:

$$v_t \overset{\text{def}}{=} \beta \cdot v_{t-1} + (1 - \beta)(e_t - e_{t-1})$$

$$= (1 - \beta) \cdot \sum_{i=0}^{t} \beta^i (e_{t-i} - e_{t-i-1}) \tag{20}$$

We then proceed by separating out the sum in (20), recalling that $e_{-1} = 0$:

$$v_t = (1 - \beta) \left[ \sum_{i=0}^{t} \beta^i \cdot e_{t-i} - \sum_{i=1}^{t} \beta^{i-1} \cdot e_{t-i} \right]$$

$$= (1 - \beta) \left[ e_t - \sum_{i=1}^{t} \beta^{i-1}(1 - \beta) \cdot e_{t-i} \right]$$

$$= (1 - \beta) \left[ e_t - \frac{1 - \beta}{\beta} \cdot e_t - \sum_{i=0}^{t} \beta^{i-1}(1 - \beta) \cdot e_{t-i} \right]$$

$$= \frac{1 - \beta}{\beta} \left[ e_t - (1 - \beta) \sum_{i=0}^{t} \beta^i \cdot e_{t-i} \right] \tag{21}$$

The desired result follows by substituting $e_t = -\nabla \hat{L}_t(\theta_t)$ into (21). $\qquad\square$

# H  QHM AND AGGREGATED MOMENTUM

We perform a brief empirical comparison of QHM and Aggregated Momentum (AggMo), proposed by Lucas et al. (2018). In short, we find that for an autoencoder task, we can take the optimal parameterization of AggMo from an extensive parameter sweep, and from that we can construct a QHM parameterization by hand which outperforms the optimal AggMo parameterization.

## H.1  ALGORITHM: AGGREGATED MOMENTUM (AGGMO)

AggMo is a many-state optimizer that aggregates multiple momentum buffers in its update rule.

**AggMo update rule**  The AggMo algorithm, parameterized by discount factors $\beta \in \mathbb{R}^K$ and learning rate $\gamma > 0$, uses the update rule:

$$g_{t+1}^{(i)} \leftarrow \beta^{(i)} \cdot g_t^{(i)} + \nabla \hat{L}_t(\theta_t) \qquad \text{for } i \in [1, K] \tag{22}$$

$$\theta_{t+1} \leftarrow \theta_t - \gamma \left[ \frac{1}{K} \cdot \sum_{i=1}^{K} g_{t+1}^{(i)} \right] \tag{23}$$

Intuitively, AggMo maintains $K$ unnormalized momentum buffers with different discount factors and uses the average of these buffers in the update rule.

## H.2  EMPIRICAL COMPARISON OF QHM AND AGGMO

**Experimental setup: EMNIST autoencoders**  We perform the autoencoder experiments of Lucas et al. (2018) using the authors' implementation, [28] with two changes:

1. We replace the MNIST dataset (LeCun, 1998) with the richer digits subset of the EMNIST dataset (Cohen et al., 2017). We hold out 10% of the training dataset for validation.

2. We change the minibatch size from 200 to 256 for computational efficiency.

**Parameterizing AggMo**  Lucas et al. (2018) conduct a sweep over parameterizations of AggMo. Performing the same sweep, we find that the best parameterization of AggMo uses discount factors $\beta = [0, 0.9, 0.99, 0.999]$ and learning rate $\gamma = 0.1$. We name this parameterization "AggMo-Best".

**Parameterizing QHM**  We now apply intuition to convert AggMo-Best into a QHM parameterization, which we name "QHM-Converted". We calculate the effective step size $\alpha$ of AggMo-Best:

$$\alpha = \frac{\gamma}{4} \cdot \sum_{i=1}^{4} \frac{1}{1 - \beta^{(i)}}$$

$$= \frac{1}{40} \left[ 1 + 10 + 100 + 1000 \right]$$

$$= 27.775$$

We round up and use $\alpha = 28$ as the learning rate for QHM-Converted.

From Section 7.1, our rule of thumb for QHM is $\nu = 0.7$ and $\beta = 0.999$. However, noting that this rule of thumb is toward replacing momentum/NAG with discount factor 0.9, and observing that the best NAG parameterization reported by Lucas et al. (2018) uses discount factor 0.99, we instead use $\nu = 0.97$ and $\beta = 0.999$ for QHM-Converted.

In summary, the parameterization of QHM-Converted is $\alpha = 28$, $\nu = 0.97$, and $\beta = 0.999$, and no optimization or parameter sweeps on this task were performed to construct this parameterization.

**Results**  Fig. 4 and Table 4 present the performance of AggMo-Best and QHM-Converted on the autoencoder task. QHM-Converted outperforms AggMo-Best on the mean squared error (MSE) metric over the training, validation, and testing datasets.

---

[28]https://github.com/AtheMathmo/AggMo

Table 4: Final performance on EMNIST autoencoder task

| Optimizer | Training MSE | Validation MSE | Testing MSE |
|---|---|---|---|
| **AggMo-Best** | 2.0823±0.0312 | 2.2895±0.0236 | 2.2827±0.0247 |
| **QHM-Converted** | **1.7330**±0.0102 | **2.0921**±0.0078 | **2.0851**±0.0071 |

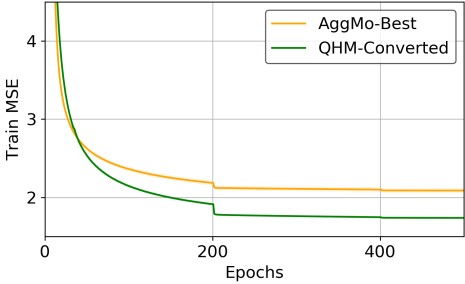 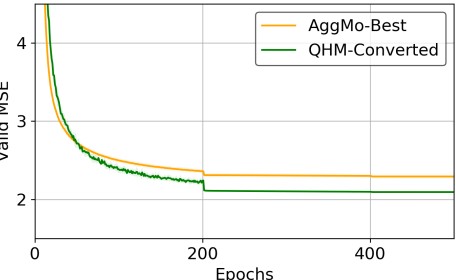

Figure 4: Training and validation MSE of AggMo-Best and QHM-Converted over the first 500 epochs. The shaded region corresponds to one standard deviation over 15 runs.

### H.3 DISCUSSION

To recap, we take the optimal AggMo parameterization from an extensive sweep, we convert that parameterization by hand to one for QHM, and we find that the latter outperforms the former on this autoencoder task.

These results indicate that using multiple momentum buffers with an arbitrary weighting scheme (i.e. AggMo with $K > 2$) provides negligible benefit over using a single slow-decaying momentum buffer with an appropriate weight (i.e. QHM with high $\beta$ and appropriate $\nu$).

**Passive damping** Lucas et al. (2018) offer an interpretation of AggMo as passive damping for physical systems. In this interpretation, fast-decaying momentum buffers "dampen" the oscillations of slow-decaying momentum buffers by providing velocity in an opposite direction.

In this context and considering these results, we conjecture that the current gradient already provides adequate damping for a slow-decaying momentum buffer, and that the damping provided by additional momentum buffers is of marginal value.

**Extended AggMo** Lucas et al. (2018) propose an extension of AggMo which allows for alternate weighting schemes via separate per-buffer learning rates. The learning rate becomes a vector $\gamma \in \mathbb{R}^K$ and (23) becomes the following:

$$\theta_{t+1} \leftarrow \theta_t - \frac{1}{K} \cdot \sum_{i=1}^{K} \gamma^{(i)} \cdot g_{t+1}^{(i)} \tag{24}$$

Lucas et al. (2018) motivate this extension by the recovery of NAG. In fact, we observe that this extension, with $K = 2$ and discount factors $[0, \beta]$, recovers QHM as well.

In independent preliminary experiments on different tasks, we found that various alternate weighting schemes of multiple momentum buffers (i.e. various parameterizations of extended AggMo with $K > 2$) did not result in material improvements over the single momentum buffer. However, this preliminary investigation was neither rigorous nor conclusive. Lucas et al. (2018) do not empirically explore these alternate weighting schemes, and it is unclear how to do so both comprehensively and efficiently, since the number of hyperparameters scales linearly with the number of momentum buffers $K$.

Toward improving the usability of extended AggMo, we suggest as future work to investigate theoretically grounded or empirically tractable methods to determine good weighting schemes for extended AggMo. However, given the added costs and complexity of AggMo (both standard and extended), we surmise in the meantime that QHM may be preferable for most practical applications.

## I  FULL DETAILS OF EXPERIMENTAL SETUP

**Environment**  All experiments use Python 3.7 and PyTorch 0.4.1 (Paszke et al., 2017). Experiments are run on a mix of NVIDIA P100 and V100 GPUs, along with a mix of CUDA 9.0 and 9.2.

### I.1  PARAMETER SWEEP EXPERIMENTS

**Common settings (all experiments)**  Training occurs over 90 epochs (minibatch size 64). The first epoch uses linear warmup of the learning rate $\alpha$ (i.e. $\alpha$ starts from zero and grows to its "regular" value by the end of the epoch). Each training run uses a single GPU.

Each parameterization is run 3 times with different seeds, and we report training loss, training top-1 error, and validation top-1 error.

**Common settings (QHM experiments)**  We use a step decay schedule for the learning rate: $\alpha \in \{1, 0.1, 0.01\}$. That is, the first 30 epochs use $\alpha = 1.0$, the next 30 epochs use $\alpha = 0.1$, and the final 30 epochs use $\alpha = 0.01$. [29]

We sweep over $\nu$ and $\beta$ using the following two-dimensional grid:

$$\nu \in \{0, 0.25, 0.5, 0.6, 0.7, 0.8, 0.9, 0.95, 0.98, 0.99, 0.995, 0.998, 0.999, 0.9995, 1\}$$
$$\beta \in \{0, 0.25, 0.5, 0.6, 0.7, 0.8, 0.9, 0.95, 0.98, 0.99, 0.995, 0.998, 0.999, 0.9995\}$$

Note that this grid encapsulates numerous parameterizations of plain SGD, momentum, and NAG (specifically, all parameterizations with the $\beta$ values enumerated above).

**Common settings (QHAdam experiments)**  We fix $\alpha = 0.001$, $\beta_2 = 0.999$, and $\epsilon = 10^{-8}$, as suggested in Kingma & Ba (2015). We also fix $\nu_2 = 1$.

We sweep over $\nu_1$ and $\beta_1$ using the same grid as for QHM's $\nu$ and $\beta$.

#### I.1.1  EXPERIMENT: LOGISTIC-EMNIST-QHM

**Model**  The model is multinomial logistic regression with pixel vector input.

**Task**  The task is digit recognition over the EMNIST dataset – specifically, the digits subset (Cohen et al., 2017).

**Optimizer**  The model is optimized with QHM. The optimization objective is cross-entropy loss, plus L2 regularization with coefficient $\frac{1}{2} \cdot 10^{-4}$.

#### I.1.2  EXPERIMENT: LOGISTIC-EMNIST-QHADAM

**Model**  Same as in Logistic-EMNIST-QHM.

**Task**  Same as in Logistic-EMNIST-QHM.

**Optimizer**  The model is optimized with QHAdam. The optimization objective is the same as in Logistic-EMNIST-QHM.

#### I.1.3  EXPERIMENT: MLP-EMNIST-QHM

**Model**  The model is a multilayer perceptron (specifically, 3 layer feed forward network) with pixel vector input. The hidden layer sizes are 200, 100, and 50 units, and all hidden units are $\tanh$ non-linearities. The final layer is followed by softmax.

**Task**  Same as in Logistic-EMNIST-QHM.

**Optimizer**  Same as in Logistic-EMNIST-QHM.

---

[29]These learning rates may seem high, but recall that the effective step size is identical to that of "typical", unnormalized momentum/NAG with $\alpha \in \{0.1, 0.01, 0.001\}$ and $\beta = 0.9$.

### I.1.4 EXPERIMENT: MLP-EMNIST-QHADAM

**Model**  Same as in MLP-EMNIST-QHM.

**Task**  Same as in MLP-EMNIST-QHM.

**Optimizer**  Same as in Logistic-EMNIST-QHAdam.

### I.1.5 EXPERIMENT: RN18-CIFAR10-QHM

**Model**  The model is a 18-layer convolutional residual network with preactivations (He et al., 2016b).

**Task**  The task is image recognition on the CIFAR-10 dataset (Krizhevsky, 2009).

**Optimizer**  The model is optimized with QHM. The optimization objective is cross-entropy loss, plus L2 regularization with coefficient $\frac{1}{2} \cdot (5 \cdot 10^{-4})$.

**Other details**  The implementation generally follows Liu (2017). Data augmentation includes horizontal flipping at random, as well as random 32-pixel crops with 4-pixel padding. For batch normalization (Ioffe & Szegedy, 2015), we use online calculation of moments with 0.99 exponential decay.

### I.1.6 EXPERIMENT: RN50-IMAGENET-QHM

**Model**  The model is a 50-layer convolutional residual network (He et al., 2016a).

**Task**  The task is image recognition on the ILSVRC2012 ("ImageNet") 1000-class dataset (Russakovsky et al., 2015).

**Optimizer**  The model is optimized with QHM. The optimization objective is cross-entropy loss, plus L2 regularization with coefficient $\frac{1}{2} \cdot 10^{-4}$.

**Other details**  The implementation generally follows Paszke et al. (2016). Data augmentation includes horizontal flipping at random, as well as random 224-pixel crops. Validation is performed on 224-pixel center crops. For batch normalization, we use online calculation of moments with 0.99 exponential decay. The model is trained with half-precision floating point.

## I.2 CASE STUDIES

### I.2.1 EXPERIMENT: RN152-IMAGENET-QHM

**Model**  The model is a 152-layer convolutional residual network (He et al., 2016a).

**Task**  Same as in RN50-ImageNet-QHM.

**Optimizer (baseline)**  We use the baseline configuration (NAG with size-256 minibatches) described in Goyal et al. (2017). Specifically, the learning rate schedule is $\alpha = 0.1$ for the first 30 epochs, $\alpha = 0.01$ for the next 30 epochs, $\alpha = 0.001$ for the next 20 epochs, and $\alpha = 0.0001$ for the final 10 epochs. The optimization objective is cross-entropy loss, plus L2 regularization with coefficient $\frac{1}{2} \cdot 10^{-4}$. The only departure from Goyal et al. (2017) is that we employ a first-epoch linear warmup of $\alpha$ (as in the parameter sweep experiments).

**Optimizer (QHM)**  The non-baseline optimizer is QHM with $\nu = 0.7$ and $\beta = 0.999$. Following Section 7.1, we increase the learning rate ($\alpha$) 10-fold. All other details are identical to the baseline.

**Evaluation**  For each optimizer, we run 3 seeds and report validation top-1 error.

**Other details**  See RN50-ImageNet-QHM for implementation details.

### I.2.2 EXPERIMENT: FCONVLM-WIKITEXT103-QHM

**Model**  The model is the GCNN-14 variant of the gated convolutional language model described in Dauphin et al. (2016).

**Dataset**  The task is language modeling on the WikiText-103 language dataset (Merity et al., 2016).

**Optimizer (baseline)** For the baseline, we use the configuration described in Dauphin et al. (2016). Specifically, the model is optimized with 60 epochs of NAG ($\beta = 0.99$). We initialize the learning rate to $\alpha = 1.0$, and we halve it when validation loss begins to increase. The optimization objective is adaptive cross-entropy, plus direct weight decay with coefficient $5 \cdot 10^{-6}$. We also clip gradient norm with maximum norm of 0.1.

**Optimizer (QHM)** The non-baseline optimizer is QHM with $\nu = 0.98$ and $\beta = 0.998$. Following Section 7.1, we increase the initial learning rate ($\alpha$) 100-fold. All other details are identical to the baseline.

**Evaluation** For each optimizer, we run 10 seeds and report validation perplexity.

**Other details** The implementation is exactly that of fairseq-py (Gehring et al., 2017). We train each model on 8 GPUs with half-precision floating point.

### I.2.3 EXPERIMENT: TD3-MUJOCO-QHADAM

**Model** We use the Twin Delayed Deep Deterministic Policy Gradients (TD3) algorithm (Fujimoto et al., 2018) for actor/critic learning. Both the actor and the critic are represented as multilayer perceptrons.

**Task** We use a suite of MuJoCo continuous control tasks (Todorov et al., 2012). In particular, we perform evaluation on the following environments: HalfCheetah, Hopper, Walker2d, Ant, Reacher, InvertedPendulum, and InvertedDoublePendulum.

**Optimizer (baseline)** For the baseline, we use Adam with the default parameters ($\alpha = 0.001$, $\beta_1 = 0.9$, $\beta_2 = 0.999$, and $\epsilon = 10^{-8}$) as in Fujimoto et al. (2018). We train for $10^6$ iterations.

**Optimizer (QHAdam)** The non-baseline optimizer is QHAdam. We set $\nu_1 = 0.9$, $\nu_2 = 1$, and otherwise leave the baseline setting unchanged.

**Evaluation** For each optimizer, we run 10 seeds. For each seed, we report average reward (on 10 episodes) every 5000 iterations of training, following Fujimoto et al. (2018).

**Other details** We use Fujimoto et al. (2018)'s open-sourced implementation [30], along with version 2 of OpenAI Gym (Brockman et al., 2016).

### I.2.4 EXPERIMENT: TF-WMT16ENDE-QHADAM

**Model** We use a Transformer-based model (Vaswani et al., 2017) described in Ott et al. (2018).

**Task** We train our models on a filtered version of the WMT16 English-German machine translation dataset as in Vaswani et al. (2017), and evaluate on *newstest14* for English-German. Our evaluation setup is identical to the one in Ott et al. (2018).

**Optimizer (baseline)** For the baseline, we use Adam ($\beta_1 = 0.9$, $\beta_2 = 0.98$, and $\epsilon = 10^{-8}$) as in Ott et al. (2018), training over 70 epochs. We use the same learning rate schedule as in Vaswani et al. (2017) and Ott et al. (2018). Specifically, we increase the learning rate from $10^{-7}$ to $10^{-3}$ linearly for 4000 steps, then decay it proportionally to the inverse square root of the number of steps. We optimize the label smoothed cross-entropy loss as in Vaswani et al. (2017), with label smoothing of 0.1.

**Optimizer (QHAdam)** We use QHAdam ($\nu_1 = 0.8$, $\beta_1 = 0.95$, $\nu_2 = 0.7$, $\beta_2 = 0.98$, and $\epsilon = 10^{-8}$) as a non-baseline optimizer. The other settings are identical to the baseline.

**Evaluation** For each optimizer, we run 10 seeds and report validation perplexity and validation BLEU scores. We observe that 4 seeds for the baseline Adam optimizer "explode" (fail to converge). Thus, we only consider the 6 best seeds for each optimizer.

**Other details** We use Ott et al. (2018)'s open-sourced implementation in fairseq-py (Gehring et al., 2017). We train each model on 8 GPUs with half-precision floating point. Note that Ott et al. (2018) uses 128 GPUs; to eliminate this discrepancy and precisely reproduce the training environment, we accumulate gradients over 16 minibatches before each optimization step.

---

[30]`https://github.com/sfujim/TD3`

## J FULL PARAMETER SWEEP RESULTS

Fig. 5 shows summary graphs for all parameter sweep experiments. These summary graphs display selected parameterizations and optimal parameterizations of both the vanilla and QH algorithms. Full details of experimental settings can be found in Appendix I.

Since each experimental setting contains nearly 200 parameterizations with 3 seeds each, we cannot fully present the data with graphs or tables. Thus, we provide data files describing all runs in CSV format. [31]

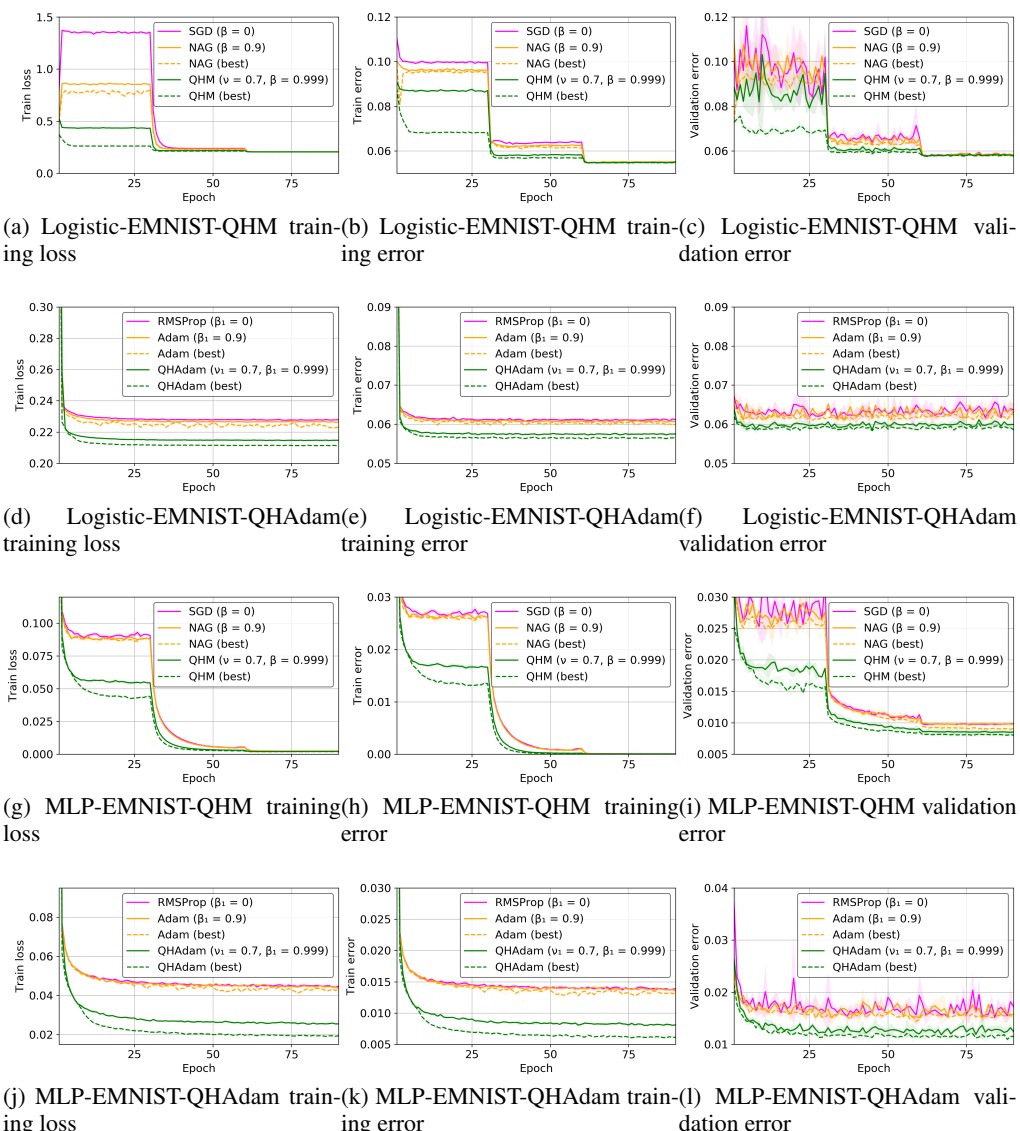

(a) Logistic-EMNIST-QHM train-ing loss

(b) Logistic-EMNIST-QHM train-ing error

(c) Logistic-EMNIST-QHM vali-dation error

(d) Logistic-EMNIST-QHAdam training loss

(e) Logistic-EMNIST-QHAdam training error

(f) Logistic-EMNIST-QHAdam validation error

(g) MLP-EMNIST-QHM training loss

(h) MLP-EMNIST-QHM training error

(i) MLP-EMNIST-QHM validation error

(j) MLP-EMNIST-QHAdam train-ing loss

(k) MLP-EMNIST-QHAdam train-ing error

(l) MLP-EMNIST-QHAdam vali-dation error

Figure 5: Full parameter sweep results (part 1 of 2). Shaded bands indicate $\pm 1$ standard deviation.

---

[31] https://github.com/facebookresearch/qhoptim/releases/download/emptytag/qhoptim_parameter_sweep_data.tar.gz

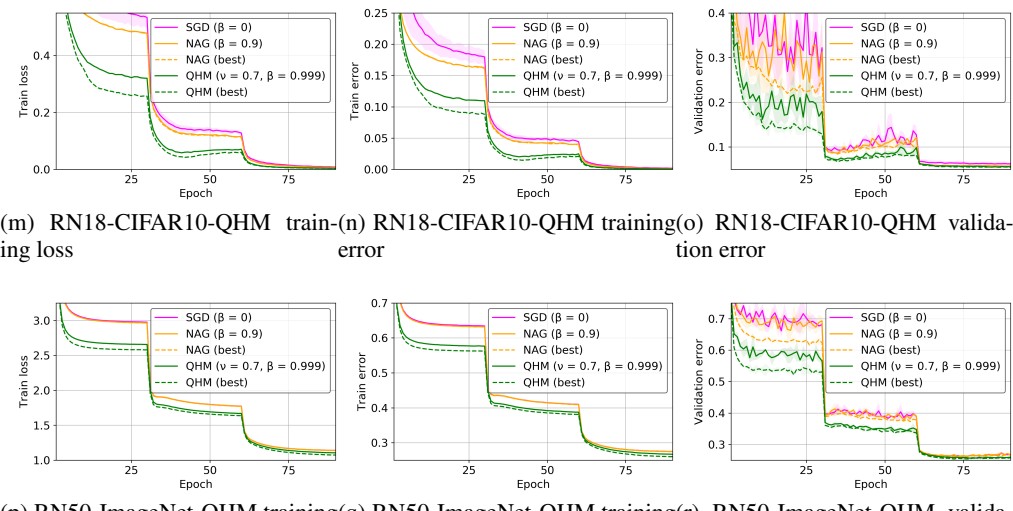

(m) RN18-CIFAR10-QHM train-
ing loss

(n) RN18-CIFAR10-QHM training
error

(o) RN18-CIFAR10-QHM valida-
tion error

(p) RN50-ImageNet-QHM training
loss

(q) RN50-ImageNet-QHM training
error

(r) RN50-ImageNet-QHM valida-
tion error

Figure 5: Full parameter sweep results (part 2 of 2). Shaded bands indicate $\pm 1$ standard deviation.

