# OpenReview forum: "Quasi-hyperbolic momentum and Adam for deep learning"
_ICLR.cc/2019/Conference_

### Official Review · AnonReviewer1 · 2018-11-02

**Rating:** 8
**Confidence:** 3

**Review:**

The authors introduce a class of quasi-hyperbolic algorithms that mix SGD with SGDM (or similar with Adam) and show improved empirical results. They also prove theoretical convergence of the methods and motivate the design well. The paper is well-written and contained the necessary references. Although I did feel that the authors could have better compared their method against the recent AggMom (Aggregated Momentum: Stability Through Passive Damping by Lucas et al.). Seems like there are a few similarities there.

I enjoyed reading this paper and endorse it for acceptance. The theoretical results presented and easy to follow and state the assumptions clearly. I appreciated the fact that the authors aimed to keep the paper self-contained in its theory. The numerical experiments are thorough and fair. The authors test  the algorithms on an extremely wide set of problems ranging from image recognition (including CIFAR and ImageNet), natural language processing (including the state-of-the-art machine translation model), and reinforcement learning (including MuJoCo). I have not seen such a wide comparison in any paper proposing training algorithms before. Further, the numerical experiments are well-designed and also fair. The hyperparameters are chosen carefully, and both training and validation errors are presented. I also appreciate that the authors made the code available during the reviewing phase. Out of curiosity, I ran the code on some of my workflows and found that there was some improvement in performance as well.

---

> ### Author Response · Authors · 2018-11-12
> **Review response -- thanks for the feedback!**
>
> We thank the reviewer for their encouraging and constructive feedback. We are heartened that the reviewer has found the algorithms useful for their own applications!
>
> # Using multiple momentum buffers
>
> We appreciate the pointer to the AggMo algorithm (Lucas et al., 2018), which proposes the additive use of many momentum buffers with different values of beta (the momentum constant). We had tried this in independent preliminary experimentation (toward analyzing many-state optimization), and we found that using multiple momentum buffers yields negligible value over using a single slow-decaying momentum buffer and setting an appropriate immediate discount (i.e. QHM with high beta and appropriate nu). Given the added costs and complexity of using multiple momentum buffers, we decided against discussing many-state optimization.
>
> We believe that the two papers are largely orthogonal, as one paper focuses in depth on two-state optimization, while the other more broadly explores many-state optimization. However, in light of AggMo's existence, we believe it is valuable to comment on the relationship between QHM and AggMo. Specifically, we have updated the manuscript as follows:
> - In section 4.5, we briefly connect QHM to AggMo.
> - In Appendix H, we provide a supplemental discussion and comparison with AggMo. Specifically, we perform the autoencoder study from Appendix D.1 of Lucas et al. (2018) with both algorithms, using the EMNIST dataset. In short, we believe that the results of this comparison support the above notion from our preliminary experimentation.

---

> > ### Comment · AnonReviewer1 · 2018-11-20
> > **Thank you for your reply**
> >
> > Thanks for your clarifications. I am retaining my rating; I maintain that this is a good paper and endorse it for publication.

---

> > > ### Author Response · Authors · 2018-11-26
> > > **Thanks for the followup!**
> > >
> > > Thanks for the follow-up, and we are glad that the reviewer enjoyed the paper!

---

### Official Review · AnonReviewer3 · 2018-11-03
**Simple idea. Impressive results. Some discussion needed to be more convincing.**

**Rating:** 6
**Confidence:** 4

**Review:**

Update after the author response: I am changing my rating from 6 to 7. The authors did a good job at clarifying where the gain might be coming from, and even though I maintain that decoupling the two variables is a simple modification, it leads to some valuable insights and good results which would of interest to the larger research community.

-------
In this paper the authors propose simple modifications to SGD and Adam, called QH-variants, that can not only recover the “parent” method but a host of other optimization tricks that are widely used in the applied deep learning community. Furthermore, the resulting method achieves better performance on a suit of different tasks making it an appealing choice over the competing methods.

Training a DNN can be tricky and substantial efforts have been made to improve on the popular SGD baseline with the goal of making training faster or reaching a better minima of the loss surface. The paper introduces a very simple modification to existing algorithms with surprisingly promising results. For example, on the face of it, QHM which is the modification of SGD, is exactly like momentum except we replace \beta in eq. 1 to \nu*\beta. Without any analysis, I am not sure how such a change leads to dramatic difference in performance like the first subfigure in Fig. 2. The authors say that the performance of SGD was similar to that of momentum, but performance of momentum with \beta = 0.7*0.999 should be the same as that of QHM. So where is the gain coming from? What am I missing here? Outside of that, the results are impressive and the simplicity of the method quite appealing. The authors put in substantial efforts to run a large number of experiments and providing a lot of extra material in the appendix for those looking to dive into all the details which is appreciated.


In summary, there are a few results that I don’t quite follow, but the rest of the paper is well organized and the method shows promise in practice. My only concern is the incremental nature of the method, which is only partly offset by the good presentation.

---

> ### Author Response · Authors · 2018-11-08
> **Review response -- thanks for the feedback!**
>
> We thank the reviewer for their encouraging and constructive feedback.
>
> # QHM vs. momentum
>
> We appreciate the reviewer raising this potential point of confusion, and we would like to emphasize that replacing beta with (nu * beta) in momentum *does not* recover QHM.
>
> Analytically, we note that replacing beta with (nu * beta) in Equation 2 propagates nu into the momentum buffer (g_t) via Equation 1, ultimately changing the decay rate of the momentum buffer from beta to (nu * beta). Intuitively, we note that QHM constitutes the *complete* decoupling of the momentum buffer's decay rate (beta) from the current gradient's contribution to the update rule (1 - nu * beta). In contrast, momentum tightly couples the decay rate (beta) and the current gradient's contribution (1 - beta).
>
> It is crucial to understand this difference as it reveals QHM's added expressivity over momentum, and we concur that more explicit discussion of this difference would be beneficial. We have updated the manuscript as follows:
> - Appendix A.8 analytically demonstrates the difference between the two, in terms of the weight on each past gradient.
> - Section 3 of the main text briefly and intuitively describes the added expressive power of QHM over momentum, in line with the above explanation.
>
> # Incrementality
>
> We appreciate the reviewer's honest assessment of the incrementality of the approach, but respectfully disagree. In the interest of accessibility, we have intentionally presented the simplest possible exposition of the algorithm, rather than the various more complex formulations possible with our original motivation. On first principles, we believe that this simplicity is a benefit rather than a disadvantage. Yet this simplicity belies both theoretical and practical power. Theoretically, we have demonstrated that many powerful but opaque optimization algorithms (essentially, all two-state linear first-order optimizers) boil down to decoupling the momentum buffer's decay rate from the current gradient's weight, and we have presented the most direct and efficient method to do so. And practically, we have demonstrated improvements that are at least as significant as the improvement between plain SGD and momentum/NAG.
>
> Although we wish to err toward understating rather than overstating our contributions, we would be deeply appreciative of any suggestions the reviewer could offer to improve the articulation of these points in the manuscript.

---

### Official Review · AnonReviewer2 · 2018-11-09
**The paper presents some interesting results but I found some of the content hard to follow**

**Rating:** 7
**Confidence:** 4

**Review:**

Edit: Following response, I have updated my score from 6 to 7.

I completed this review as an emergency reviewer - meaning that I had little time to complete the review. I did not have time to cover all of the material in the lengthy appendix but hope that I explored the parts most relevant to my comments below.

Paper summary: The paper introduces QHM, a simple variant of classical momentum which takes a weighted average of the momentum and gradient update. The authors comprehensively analyze the relationships between QHM and other momentum based optimization schemes. The authors present an empirical evaluation of QHM and QHAdam showing comparable performance with existing approaches.

Detailed comments:

I'll use CM to denote classical momentum, referred to as "momentum" in the paper.


1) In the introduction, you reference gradient variance reduction as a motivation for QHM. But in Section 3 you defer readers to the appendix for the motivation of QHM. I think that the main paper should include a brief explanation of this motivation.

2) The proposed QHM looks quite similar to a special case of Aggregated Momentum [1]. It seems that the key difference is with the use of damping but I suspect that this can be largely eliminated by using different learning rates for each velocity (as in Section 4 of [1]) and/or adopting damping in AggMo. In fact, Section 4.1 in your paper recovers Nesterov momentum in a very similar way. More generally, could one think of AggMo as a generalization of QHM? It averages plain SGD and several momentum steps on different time scales.

3) I thought that some of the surprising relations to other momentum based optimizers was the most interesting part of the paper. However, I found the presentation a little difficult. There are many algorithms presented but none are explored fully in the main paper. I had to flick between the main paper and appendix to uncover the information I wanted most from the paper.

Moreover, I found some of the arguments in the appendix a little tough to follow. For example, with AccSGD you should specify that epsilon is a constant typically chosen to be 0.7.  When the correspondence to QHM is presented it is not obvious that QHM -> AccSGD but not the other way around. I would suggest that you present a few algorithms in greater detail, and list the other algorithms you explore at the end of Section 4 with pointers to the appendix.

4) I am not sure that the QHAdam algorithm adds much to the paper. It is not explored theoretically and I found the empirical analysis fairly limited.

5) In general, the empirical results support QHM as an improvement on SGD/NAG. But I have some (fairly minor) concerns.

   a) For Figure 1, it looks like QHM beats QHAdam on MLP-EMNIST. Why not show these on the same plot? This goes back to my point 4 - it does not look like QHAdam improves on QHM and so I am not sure why it is included. The idea of averaging gradients and momentum is general - why explore QHAdam in particular?

   b) For Figure 2, while I certainly appreciate the inclusion of error bars, they suggest that the performance of all methods are very similar. In Table 3, QH and the baselines are often not just within a standard deviation of eachother but also have very close means (relatively).

6) I feel that some of the claims made in the paper are a little strong. E.g. "our algorithms lead to significantly improved training in a variety of settings". I felt that the evidence for this was lacking.


Overall, I felt that the paper offered many interesting results but clarity could be improved. I have some questions about the empirical results but felt that the overall story was strong. I hope that the issues I presented above can be easily addressed by the authors.


Minor comments:

- I thought the use of bold text in the introduction was unnecessary
- Some summary of the less common tasks in Table 2 should be given in the main paper


Clarity: I found the paper quite difficult to follow in places and found myself bouncing around the appendix frequently. While the writing is good I think that some light restructuring would improve the flow.

Significance: The paper presents a simple tweak to classical momentum but takes care to identify its relation to existing algorithms. The empirical results are not overwhelming but at least show QHM as competitive with CM on tasks and architecture where SGD is typically dominant.

Originality: To my knowledge, the paper presents original findings and places itself well amongst existing work.


References:

[1] Lucas et al. "Aggregated Momentum: Stability Through Passive Damping" https://arxiv.org/pdf/1804.00325.pdf

---

> ### Author Response · Authors · 2018-11-12
> **Review response -- thanks for the feedback! [Part 1 of 2]**
>
> We thank the reviewer for their encouraging and constructive feedback.
>
> The reviewer has offered a large number of insightful comments, which is particularly appreciated given the exigence of the review request. For convenience, we address them by number:
>
> # 1
>
> We concur with the reviewer's suggestion and have updated Section 3 of the manuscript to provide this brief summary.
>
> # 2
>
> We appreciate the pointer to the AggMo algorithm (Lucas et al., 2018), which proposes the additive use of many momentum buffers with different values of beta (the momentum constant). We had tried this in independent preliminary experimentation (toward analyzing many-state optimization), and we found that using multiple momentum buffers yields negligible value over using a single slow-decaying momentum buffer and setting an appropriate immediate discount (i.e. QHM with high beta and appropriate nu). Given the added costs and complexity of using multiple momentum buffers, we decided against discussing many-state optimization.
>
> We believe that the two papers are largely orthogonal, as one paper focuses in depth on two-state optimization, while the other more broadly explores many-state optimization. However, in light of AggMo's existence, we believe it is valuable to comment on the relationship between QHM and AggMo. Specifically, we have updated the manuscript as follows:
> - In section 4.5, we briefly connect QHM to AggMo.
> - In Appendix H, we provide a supplemental discussion and comparison with AggMo. Specifically, we perform the autoencoder study from Appendix D.1 of Lucas et al. (2018) with both algorithms, using the EMNIST dataset. In short, we believe that the results of this comparison support the above notion from our preliminary experimentation.
>
> # 3
>
> We appreciate the feedback on the presentation of Section 4. We have attempted to cater to a diverse audience across the practitioner-theorist spectrum, and the strongest feedback we received pre-submission is that many readers on both ends of the spectrum appreciate to have in the main text only:
> - The analytical form (i.e. update rule) of the discussed algorithm, and brief efficiency discussion
> - The succinct “upshot” as it relates to QHM (i.e. narrative summary of the recovery result)
>
> and for the mathematical derivations and specific recovery parameterizations to be relegated to the appendix. In particular, we have received feedback that the matrix machinery required for most of the recoveries detracts from the main text, and any detailed derivations depend on this machinery.
>
> In recognition of the reviewer's concerns, we have updated Appendix C of the manuscript to be more structured and self-contained (essentially, a more detailed version of Sections 4.2 through 4.4), so that the more theory-minded audience might have an easier time reading without having to switch back-and-forth between Appendix C and the main text.
>
> We would very much welcome suggestions on what specific facts merit inclusion in the main paper, besides the analytical forms of the update rules and narrative relation to QHM.
>
> Regarding AccSGD specifically, we have updated the manuscript to more clearly explain the one-way nonrecovery (both in the main text and in the appendix). We believe that our current method of showing this nonrecovery (via NAG) is the most accessible, while revealing a useful erratum in the prior work of Kidambi et al. (2018).

---

> > ### Author Response · Authors · 2018-11-12
> > **Review response -- thanks for the feedback! [Part 2 of 2]**
> >
> >
> > # 4 & 5
> >
> > We acknowledge that formal convergence analysis is not provided for QHAdam. Nevertheless, we believe that the contradiction of the widely-accepted Adam step size bound from Kingma & Ba (2015) and QHAdam's theoretically grounded ability to tighten this bound is of substantial interest. We believe that we have indeed demonstrated the empirical usefulness of this with the NMT case study. Increasing from 60% to 100% robustness is a large improvement, and an increase of 0.3 BLEU from an optimizer change alone is viewed as fairly significant in the NMT community.
> >
> > With regards to the EMNIST classification parameter sweeps, we seek to compare our algorithms with their own vanilla counterparts (i.e. QHAdam > Adam), without meticulously tuning the QHAdam and QHM curves to look comparable with one another. We note that there is a certain non-standard LR schedule for (QH)Adam which surpasses the results shown for QHM. However, for the purposes of this study, we believe it best to stick to the standard Adam LR. More generally, we lament the trend of comparing adaptive and non-adaptive methods side-by-side when the terms of comparison are questionable at best. Fair comparison of adaptive and non-adaptive methods is likely a suitable subject for an entirely new paper.
> >
> > Finally, we wish to make a broader point regarding the “case study” experiments. Our primary goal in performing these case studies is to demonstrate practically realistic scenarios. Thus, we did not perform systematic sweeps to squeeze all possible performance out of the algorithms. Rather, we approached the case studies as we felt a practitioner would, relying on intuition to translate the vanilla optimizer to the QH optimizer. In that light, we believe that the case study results as a whole are compelling:
> > - We observe *much* faster convergence in image recognition and marginal/neutral results in final validation accuracy. In general, one should not expect significant differences in final validation accuracy on the standard ResNet+ImageNet combo, assuming that the optimizer has trained the model to convergence.
> > - We observe respectably lower perplexity in language modeling. Note that though the SD bars overlap here, the results are still statistically significant (at the 0.1% confidence level) since using 10 seeds results in a reduced standard error.
> > - We observe neutral results in reinforcement learning.
> > - We observe notable robustness and performance improvements in NMT, as discussed above. The graph is primarily for illustrative purposes, since the metric of interest is BLEU (which is only highlighted in the table).
> >
> > # 6/Overall
> >
> > We hope that our updates to the manuscript address the reviewer's concerns about clarity, and we hope that the discussion above addresses the reviewer's concerns about empirical significance. We once again thank the reviewer for the incredibly thorough commentary of our manuscript.

---

> > > ### Comment · AnonReviewer2 · 2018-11-16
> > > **Thank you for your response. Review score updated.**
> > >
> > > #1 This looks good!
> > >
> > > #2 I think that the new additions to the paper do a great job of distinguishing QHM and AggMo while exposing their similarities. I am not sure that I agree with the two works being entirely orthogonal, but I think that the revision is more than fair in its comparison of the two.
> > >
> > > #3 I understand what you are saying. While you should weigh your presentation against the opinions of the many, as a reviewer it is my job to give feedback from my position. I still believe that the main paper struggles from some of the issues I presented in my initial review. However, the appendix does seem easier to read and the additions to the AccSGD section are good. Though we are left in disagreement, I think overall that my issue is a minor point which has been addressed to some extent.
> > >
> > > I don't have any specific recommendations beyond what I said in my initial review. However, I respect that my bias may be in conflict with other feedback you have received.
> > >
> > > #4 & 5 In my initial review I did not have time to explore Appendix F. I must confess that I still have not been able to cover all of the details. However, I am still not completely convinced by some aspects of QHAdam. In particular, some of the theoretical arguments in the appendix. Disproving the step size bound seems interesting, though I do not entirely understand the significance. It seems the key theoretical argument for QHAdam over Adam is the ability to recover a tighter step size bound. Perhaps this should be made clear in the main text (expanding on "it is possible that setting v_2 < 1 can improve stability"). Moreover, why is this method of reducing the step size more effective than simple reducing beta_2 in Adam? You claim that small beta_2 values can lead to slow convergence, how does reducing v_2 instead correct this?
> > >
> > > Thank you for clarifying the empirical results. After taking a more careful look, I agree that QHAdam seems worthwhile to include. I am not familiar with NMT optimization, is the idea of a spiky gradient distribution well established? While I acknowledge QHAdam gives a significant win on this task, I am not yet convinced by the proposed explanation. However, I do not see this as a critical component of the paper.
> > >
> > > To summarize, with your explanation here I am more convinced by the empirical results than on my first reading.
> > >
> > > #6 It is impossible to get a second first-impression, but I feel that in general the clarity has been improved.
> > >
> > > - Minor point: Why relegate Fact F1 proof to appendix G?
> > >
> > > Thank you for addressing the points I raised. After reading your response, I am more convinced that the paper should be accepted and have thus increased my original score from 6 to 7.

---

> > > > ### Author Response · Authors · 2018-11-26
> > > > **Thanks for revisiting the assessment! Response to remaining concerns.**
> > > >
> > > > We thank the reviewer for their generous revisiting of their assessment! Our latest update to the manuscript addresses the reviewer's remaining concerns as follows:
> > > >
> > > > - We have explicitly stated in Section 5 of the main text that the stability properties of QHAdam discussed come from the tighter step size bound.
> > > > - We have briefly elaborated on the need for large beta_2 in Appendix F.
> > > > - We have moved the proof of Fact F.1 inline.

---

### Public Comment · (anonymous) · 2018-11-07
**Question about eqn (3) and (4)?**

Hi,

I'm confused by the update rule of QHM. What's the difference between QHM and plain momentum method? From my perspective, we can rewrite eqn (3) and (4) with eqn (1) and (2) but change *beta* to *v beta*. If so, what's the advantage of QHM as we can always tune *beta*.

---

> ### Author Response · Authors · 2018-11-08
> **Thanks for the interest in our paper!**
>
> Thanks for the interest in our paper!
>
> In short, momentum cannot recover QHM via this rewriting. Please refer to the discussion thread under AnonReviewer3 for further details.

---

### Public Comment · (anonymous) · 2018-11-12
**About Convergence Analysis of QHM for Deep Learning Problems**

Dear Authors:
                 Thank you for presenting an interesting work on the optimization of deep learning problems. Could you please provide the convergence analysis of your proposed QHM in the nonconvex deep learning setting? This is because as for the SGD related methods, the convergence seems to be proved in the convex case, like Adam. Thank you very much.
                  Sincerely yours

---

> ### Author Response · Authors · 2018-11-12
> **Thanks for the interest in our paper!**
>
> Thanks for the interest in our paper!
>
> We are not aware of any compelling convergence results for gradient descent and momentum (and other common algorithms) in a general non-convex setting — the best one can do is critical point convergence.
>
> As QHM is a simple interpolation between the two, QHM similarly does not have any compelling convergence results in a general non-convex setting.

---

> > ### Public Comment · (anonymous) · 2018-11-12
> > **About Critical Point Convergence**
> >
> > Thank you for the feedback. I appreciate it. I am interested in the global convergence of critical points.  Could you then recommend some literature of critical point convergence of SGD in the nonconvex setting? I have gone through most SGD papers but did not find any literature related to this field. Thank you.

---

> > > ### Author Response · Authors · 2018-11-13
> > > **Critical point convergence for GD methods**
> > >
> > > Demonstrations of critical point convergence for GD methods (in the general smooth+non-convex setting) are most likely absent from recent literature. We recommend various online course materials, such as [1] and [2].
> > >
> > > Of course, there are various restrictions one can impose for non-convex settings that will yield more interesting results (e.g. convergence to a local optimum with known rate) -- this is the focus of much recent literature! As a sampler, you might check out [3] and [4].
> > >
> > > [1] D. Papailiopoulos, http://papail.io/teaching/901/scribe_09.pdf
> > > [2] C. Sa, http://www.cs.cornell.edu/courses/cs6787/2017fa/Lecture7.pdf
> > > [3] Ge et al., https://arxiv.org/abs/1503.02101
> > > [4] Lee et al., https://arxiv.org/abs/1602.04915

---

> > > > ### Public Comment · (anonymous) · 2018-11-13
> > > > **Thank you for materials, About the comparison between  QHMAdam and Adam.**
> > > >
> > > > Dear Authors:
> > > >                   Thank you very much for providing useful learning materials. I really appreciate it.  One question is about the comparison between QHMAdam and Adam. You have conducted various experiments to illustrate the effectiveness of QHMAdam. Some figures(e.g. Figure 1) show that the QHMAdam outperformed Adam. But the Adam did not appear in other figures(e.g. Figure 5).  Could you please explain what are the advantages of QHMAdam over Adam? Thank you very much.

---

> > > > > ### Author Response · Authors · 2018-11-13
> > > > > **QHAdam**
> > > > >
> > > > > Firstly, QHAdam is in Figure 5 -- specifically, (d), (e), (f), (j), (k), (l).
> > > > >
> > > > > There is an intuitive advantage and a theoretically grounded advantage.
> > > > >
> > > > > The intuitive advantage is that whatever benefits interpolation provides for non-adaptive methods (i.e. all the theoretical results for QHM) translate to adaptive methods. This is strictly intuitive for now -- we do not provide any theoretical demonstrations of accelerated QHAdam convergence, only empirical results.
> > > > >
> > > > > The theoretically grounded advantage is stability. Adam's updates to the parameters can be much larger than can be dealt with during training. In fact, they can be much larger than previously believed -- in our manuscript, we disprove the step size bound claimed in [5] (the original Adam paper), which had been taken as fact in subsequent literature. QHAdam offers a way to mitigate this without simply cutting the learning rate and thus making training slower; this is discussed in much theoretical depth in Appendix F, and empirically validated primarily by the NMT case study.
> > > > >
> > > > > [5] Kingma & Ba, https://arxiv.org/abs/1412.6980

---

> > > > > > ### Public Comment · (anonymous) · 2018-11-13
> > > > > > **About SGD and Alternating Minimization**
> > > > > >
> > > > > > Dear Authors:
> > > > > >        Thank you for illustration. I need to read Appendix F in detail before making comments.  As for Figure 5, I just wonder why Adam disappears in some of sub-figures like  Figure 5(a),(b) and (c). Did Adam outperform QHAdam in these figures or some other reasons? I am just curious about it. One more recommendation is that authors can also show the performance of the QHMAdam on the test data. The reason is that the Adam works well in training data, but may generalize poorly on the test set. See Figure 2 in [1].
> > > > > >          Let us explore a bit further. As you know, SGD is a dominant method for deep learning. However,  recently, alternating minimization(AM) is also attracting researchers' interest because AM can avoid gradient explosion and provide convergence guarantees[2][3]. It is easy to implement AM in parallel, and it allows for non-differentiable activation functions like Relu. AM includes the Alternating Direction Method of Multipliers(ADMM) and Block Coordinate Descent(BCD). What is your opinion on the comparison between SGD and AM?
> > > > > >           Finally, thank you again for patient explanation and hope this paper will be accepted in the ICLR conference.
> > > > > >            Sincerely yours
> > > > > > [1]  Zhang, Guoqiang, and W. Bastiaan Kleijn. "Training Deep Neural Networks via Optimization Over Graphs." 2018 IEEE International Conference on Acoustics, Speech and Signal Processing (ICASSP). IEEE, 2018. https://arxiv.org/pdf/1702.03380.pdf
> > > > > > [2]  Taylor, Gavin, et al. "Training neural networks without gradients: A scalable admm approach." International Conference on Machine Learning. 2016.
> > > > > > [3]  Global Convergence in Deep Learning with Variable Splitting via the Kurdyka-Łojasiewicz Property. https://arxiv.org/abs/1803.00225

---

> > > > > > > ### Author Response · Authors · 2018-11-13
> > > > > > > **adaptive and AM methods**
> > > > > > >
> > > > > > > We are aware of the observed poor generalization ability of Adam, and we note that quite a few manuscripts submitted to this conference seek to address this issue. This issue is out of scope for our manuscript, but we note that our results extend beyond the training dataset, as depicted by the figures.
> > > > > > >
> > > > > > > We note that Figs 5abc and 5def use identical settings, as do 5ghi and 5jkl. For our rationale for not showing them side-by-side, please refer to our response to AnonReviewer2.
> > > > > > >
> > > > > > > We (the authors) are not qualified to intelligently comment on or compare to AM methods, as we are only familiar in passing with the relevant modern literature. We suspect that you are in a much better position to speak to your question :)

---

### Meta-Review · Area_Chair1 · 2018-12-06
**simple but useful extension of NAG, with good discussion of related work**

**Confidence:** 5
**Recommendation:** Accept (Poster)

**Metareview:**

This paper presents quasi-hyperbolic momentum, a generalization of Nesterov Accelerated Gradient. The method can be seen as adding an additional hyperparameter to NAG corresponding to the weighting of the direct gradient term in the update. The contribution is pretty simple, but the paper has good discussion of the relationships with other momentum methods, careful theoretical analysis, and fairly strong experimental results. All the reviewers believe this is a strong paper and should be accepted, and I concur.